# CURATION LEAKS: MEMBERSHIP INFERENCE ATTACKS AGAINST DATA CURATION FOR MACHINE LEARNING

**Dariush Wahdany**[1]    **Matthew Jagielski**[2]    **Adam Dziedzic**[1]    **Franziska Boenisch**[1]

[1]CISPA Helmholtz Center for Information Security    [2]Anthropic

## ABSTRACT

In machine learning, curation is used to select the most valuable data for improving both model accuracy and computational efficiency. Recently, curation has also been explored as a solution for private machine learning: rather than training directly on sensitive data, which is known to leak information through model predictions, the private data is used only to guide the selection of useful public data. The resulting model is then trained solely on curated public data. It is tempting to assume that such a model is privacy-preserving because it has never seen the private data. Yet, we show that without further protection, curation pipelines can still leak private information. Specifically, we introduce novel attacks against popular curation methods, targeting every major step: the computation of curation scores, the selection of the curated subset, and the final trained model. We demonstrate that each stage reveals information about the private dataset and that even models trained exclusively on curated public data leak membership information about the private data that guided curation. These findings highlight the previously overlooked inherent privacy risks of data curation and show that privacy assessment must extend beyond the training procedure to include the data selection process. Our differentially private adaptations of curation methods effectively mitigate leakage, indicating that formal privacy guarantees for curation are a promising direction.

## 1 INTRODUCTION

Data curation has become an important part of modern machine learning (ML) pipelines (Maini et al., 2024; Wu et al., 2024), offering a principled way to select high-value data in order to maximize model performance and computational efficiency. By filtering out noisy, low-quality, or redundant samples (Gadre et al., 2023; Li et al., 2024a; Gu et al., 2025; Thrush et al., 2025), curation allows to train on the most informative points, thus improving generalization and resource utilization.

This paradigm is also particularly appealing for sensitive domains, such as finance or healthcare (Schäfer et al., 2024; Assefa et al., 2021), where the available training datasets are usually limited, which hinders the training of powerful ML models. In these settings, curation offers a key advantage: it enables model developers to leverage publicly available data pools and select a subset from these that is most relevant to their target application. Typically, the small sensitive in-domain target dataset, which represents the actual distribution of interest, or the downstream data the model is expected to perform well on, is used to guide this selection. Various techniques have been proposed to perform such guidance: for example, identifying public samples that are most similar to the target data in feature space (*embedding-based curation*) (Gadre et al., 2023; Yu et al., 2024), scoring public samples based on how much they improve accuracy on the target set (Thrush et al., 2025), or maximizing a data attribution or influence metric (*gradient-based curation*) (Park et al., 2023; Engstrom et al., 2024). We focus on **Image-based** (Gadre et al., 2023) and **TRAK-based** (Tracing with the Randomly-projected After Kernel) (Park et al., 2023) methods as well-studied representatives of embedding-based and gradient-based curation approaches, respectively.

The curated public dataset is finally used to train an ML model that outperforms models trained on *all* public data or achieves similar results with greater computational efficiency on target domain tasks. Importantly, the resulting model is never directly exposed to the sensitive in-domain target dataset.

Due to these advantages, curation is widely used in practice. Curated datasets are routinely released to the public (Penedo et al., 2023; Li et al., 2024b; Penedo et al., 2024), and in some cases, even the intermediate quality scores are made available (Together Computer, 2023). Furthermore, there exists a growing market of data curation as a service, where datasets, scores, and subsets are exchanged between organizations (DatalogyAI; Snorkel; ScaleAI). Yet, the privacy risks for the target data under such practices are, to date, not well understood. To address this gap, in this paper, we provide the first systematic study of privacy risks in curation pipelines. Therefore, we design and carry out custom membership inference attacks (Shokri et al., 2017; Carlini et al., 2022a) on data points from the target set at every stage of the pipeline, and demonstrate that each step can leak private information.

Concretely, we design and evaluate attacks against 1) the curation methods' released scores, 2) the selected public subsets, and 3) the final trained model, as shown in Figure 1. To attack curation scores, we employ Likelihood Ratio Attack (LiRA) (Carlini et al., 2022a) with signal filtering (Jagielski et al., 2023) and custom attacks based on voting schemes. We show that Image-based curation's

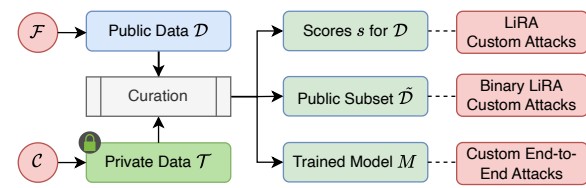

Figure 1: **We attack private data $\mathcal{T}$ used to curate a public dataset $\mathcal{D}$.** We show that the scores $s$, top-scoring subsets $\tilde{\mathcal{D}}$ and even trained models $M$ leak membership information.

nearest-neighbor mechanism creates high vulnerability for samples with non-zero influence, while TRAK's averaging provides better protection but remains vulnerable for small target datasets. To attack the curated public subset, we adapt LiRA (which typically operates on continuous model outputs) to the binary setting, where the adversary only knows if a sample was picked by the curation or not. Additionally, we design a custom attack that iteratively performs membership inference by exploiting the deterministic behavior of the curation mechanism. Finally, to attack the trained model, we insert a few *fingerprinted* samples $\mathcal{F}$ into the public pool whose curation score is highly influenced by a particular target, and which imprint a measurable signal in the model if trained on. The addition of such samples represents a realistic setup where the public pool data is often scraped from the internet, and Carlini et al. (2024) have shown that it is possible to introduce targetly manipulated data points into ML model training pipelines this way. Because our attacks succeed with only a small number of inserted samples, they hold the potential to pose a risk in real-world curation settings where public data is reused. Additionally, we explore differential privacy as a defense and show that it can mitigate these risks, though a thorough investigation of the privacy-utility trade-off remains an important direction for future work. Overall, our findings underscore that curation-guided approaches demand privacy-aware design to prevent leakage of private target data information.

To summarize, we make the following main contributions:

1. We present the first comprehensive privacy analysis of data curation pipelines, highlighting that curation leaks private information at each step: the scores, curation sets, and the final model.

2. We design custom membership attacks for each curation step, showing that both curation scores and curated datasets leak membership information without pipeline modifications.

3. We show that even end-to-end attacks on the final model can leak target data information by inserting only a small number of crafted samples into public datasets.

4. Our empirical evaluation on six datasets and two curation methods shows that, while TRAK is more robust than image-based curation, it remains highly vulnerable for small curation target datasets, the very scenario motivating curation in sensitive domains.

## 2 BACKGROUND AND RELATED WORK

**Data Attribution.** The influence of training samples on the behaviour of a trained ML model on some target data point is determined via data attribution methods. The *DataModels* approach (Ilyas et al., 2022) trains many models on different training data subsets and then fits an influence estimator. TRAK (Park et al., 2023) peforms data attribution using closed-form influence functions for Logistic Regression. To make that applicable to deep learning, they formulate the model training as a

logistic regression on gradients of a trained model. With that approach, TRAK is more efficient than DataModels, but still requires training in the order of $\sim 20$ models and another backward pass for each on the full training set. Ilyas & Engstrom (2025) introduce an alternative which requires training just one model, but requires a full backward pass on the entire training dataset *for each target sample*. By selecting those training samples with the highest positive attribution scores, data attribution methods can serve as methods for data curation (Engstrom et al., 2024).

**Data Curation.** The goal of data curation is to select the most valuable data for training. Gadre et al. (2023) introduced *DataComp*, an ML challenge of obtaining the highest accuracy just by modifying the training data. They also developed image-based filtering for selecting appropriate images based on semantic similarity. Thrush et al. (2025) propose attributing training data performance to target utility by computing correlations on the loss of the training data and the target utility. Notably, this allows computing correlations for data that the models have not been trained on, sparing the computationally expensive setup that *e.g.,* TRAK- or DataModels-based curation requires.

In this work, we focus on two representative curation methods, namely **image embedding-based curation** (Gadre et al., 2023) and **TRAK** (Park et al., 2023). In both cases, we have a public dataset $\mathcal{D} = \{x_i\}_{i=1}^N$ and a private dataset $\mathcal{T} = \{t_i\}_{i=1}^n$. We call *curating $\mathcal{D}$ for $\mathcal{T}$* when we try to obtain a subset $\tilde{\mathcal{D}} \subseteq \mathcal{D}$ s.t. the performance and/or compute efficieny for utility on $\mathcal{T}$ is improved by training on $\tilde{\mathcal{D}}$ instead of the full $\mathcal{D}$.

**Image Embedding-based Curation.** Image embedding-based curation assigns scores based on the cosine similarity of image-embeddings. For each image $x_i$ in the public curation pool $\mathcal{D}$, the score is the maximum similarity to a sample $t_j$ in the private target set $\mathcal{T}$, *i.e.,* $s(x_i) = \max_{j \in n} \cos(\phi(x_i), \phi(t_j))$ where $\phi(\cdot)$ is the embedding function.

**TRAK.** TRAK computes attribution scores via projected gradients to identify influential pool samples. Following Engstrom et al. (2024), in this work, we compute our scores as the *average attribution score* on $\mathcal{T}$, *i.e.,* $s(x_i) = \frac{1}{n} \sum_j^n \Phi_i^T G_j$ where $G_{\mathcal{T}}$ are the gradients of $\mathcal{T}$. $\Phi$ are the TRAK features. We obtain them from the gradients $X$ of $\mathcal{D}$ as $\Phi = X(X^T X)^{-1} Q$. $Q$ are the scaling factors, determined as the gradient of model output to loss.

**Assessing Privacy Risks in ML.** The de facto standard for assessing the privacy risks in ML systems is to rely on Membership Inference Attacks (MIAs) (Shokri et al., 2017; Carlini et al., 2022a). A popular attack to perform membership inference is LiRA (Carlini et al., 2022a). It trains *shadow models* on various subsets of the target data and then fits distributions to the observed behaviour of the models with a target vs. those without a target. Then, to attack a model, it compares the likelihoods of the observed behaviour under the distribution of the target being in the training vs. being not in the training data. The ratio of these likelihoods serves as the membership score. In this work, we adapt LiRA, but instead of training shadow models, we perform curation on random subsets of the target data to obtain *shadow sets*, *i.e.,* the curation results for the random subsets. We then select high-signal measurements similar to Jagielski et al. (2023), *i.e.,* those that differ most significantly for a specific target, and compute the membership inference scores as the log-likelihood ratios (Section 3.2).

## 3 ATTACKING DATA CURATION PIPELINES

In this section, we explore privacy leakage from the different stages of data curation pipelines outlined in Figure 1. We start off by outlining the threat model and adversary goal and capabilities in Section 3.1. Then, we attack three progressively harder threat models: (1) continuous curation scores (Section 3.3), (2) binary selection masks (Section 3.4), and (3) final trained models (Section 3.5). Finally, we summarize our attacks in Section 3.6.

### 3.1 THREAT MODEL

**Adversary Goal.** Our attacks infer membership in the *private target set* $\mathcal{T}_{\text{sel}} \subseteq \mathcal{T}$ used for curation, *i.e.,* for a given target $t$ whether $t \in \mathcal{T}_{\text{Sel.}}$. A target sample $t$ can influence the curation scores $s$, and through that, subsequently, the curated public dataset $\tilde{\mathcal{D}}$ and, finally, the model $\mathcal{M}$ trained on $\tilde{\mathcal{D}}$. Unlike in classical MIA (Shokri et al., 2017; Carlini et al., 2022a) $t$ is never part of the training data of the model $\mathcal{M}$ directly, *i.e.,* $\tilde{\mathcal{D}} \cap \mathcal{T} = \emptyset$.

| Attack Surface | Goal | Knowledge | Capabilities | Observations |
|---|---|---|---|---|
| **Scores** (Section 3.3) | | | Passive observation | Scores $s \in \mathbb{R}^{|\mathcal{D}|}$ |
| **Subset** (Section 3.4) | Infer $\mathcal{T}_{\text{Sel.}} \subseteq \mathcal{T}$ | Public pool $\mathcal{D}$, target set $\mathcal{T}$, curation algorithm | Passive observation | Selection $m \in \{0,1\}^{|\mathcal{D}|}$ |
| **Final Model** (Section 3.5) | | | **Inject** fingerprinted samples $\mathcal{F}$ into $\mathcal{D}$ before curation | Trained model $\mathcal{M}$ (black-box query access) |

Table 1: **Threat model summary.** All adversaries aim to infer membership in the private curation target set $\mathcal{T}_{\text{sel}}$ (*not* training-set membership). Scores provide fine-grained ranking information, subsets reveal only binary selection, and final models require active poisoning with detectable fingerprints.

**Adversary Capabilities and Knowledge.** Across all pipeline stages, we assume the following **adversary knowledge**; (1) The full public pool $\mathcal{D}$. Since such pools are often web-scale dataset from the internet, we assume the adversary can obtain it. (2) The target dataset $\mathcal{T}$. This follows standard assumptions in MIAs (Shokri et al., 2017) where the adversary knows the target samples whose membership they want to infer. (3) The curation algorithm used (*e.g.,* Image-based or TRAK). Curation methods are often open-source or disclosed in model documentation. For **adversary capabilities** we assume the adversary can observe only the outcome of the respective curation stage, *i.e.,* the scores $s$ or the selection mask $m$ or (through black-box query access) the trained model $\mathcal{M}$. Only for attacks on the final models we assume a small part of the public pool can be poisoned, which Carlini et al. (2024) have shown to be realistic. Table 1 summarizes our threat model.

## 3.2 ADAPTING LiRA FOR CURATION

LiRA (Carlini et al., 2022a) is a theoretically grounded membership inference test based on likelihood ratios. We adapt it as our principled baseline across all curation stages. The key adaptation is to replace LiRA's shadow *models* with shadow *curation sets*. Concretely, we sample $m$ different random subsets from the target dataset $\mathcal{T}$ and perform curation based on each of them. Each target is in exactly half of the random subsets, ensuring an unbiased estimate of the in/out distributions. The resulting *shadow sets* $\{s_i \in \mathbb{R}^N\}_{j=1}^m$ with $N = |\mathcal{D}|$ of curation outputs take the role of the shadow models in the original LiRA (see Section B.1 for details). For each shadow set, the adversary has ground truth membership information.

A key challenge is that the public pool is typically very large (*e.g.,* $N \approx 12.8$M for CommonPool), so each target's membership signal is diluted across millions of curation outputs. Inspired by Jagielski et al. (2023), who address an analogous signal-dilution problem for language model distillation, we select only the single most informative public sample per target, namely the one whose curation output shows the largest difference between the member and non-member distributions:

$$k^* = \arg \max_{k \in [N]} \left| \mathbb{E}_{j:t \in \mathcal{T}_j}[s_j^{(k)}] - \mathbb{E}_{j:t \notin \mathcal{T}_j}[s_j^{(k)}] \right|. \tag{1}$$

Given $k^*$, LiRA computes a membership inference score $v(t)$ as a log-likelihood ratio over the in/out distributions of the selected sample's curation output. The exact distributional form depends on the observation type—continuous scores (Section 3.3) vs. binary selection masks (Section 3.4)—and is detailed in the respective sections.

All our attacks produce a scalar membership inference score $v(t)$ per target $t$; higher values indicate stronger evidence of membership. Beyond the principled LiRA adaptations, we also design custom attacks that more targetedly exploit specific curation algorithm characteristics.

## 3.3 SCORE-BASED ATTACKS

We first analyze the privacy risks that arise from access to continuous curation scores.

**LiRA for Scores.** Using the shadow sets from Section 3.2, we observe continuous score vectors. For the selected public sample $k^*(t)$, we fit Gaussian distributions to the in-member and out-member score populations and compute the membership inference score as

$$v_{\text{LiRA}}(t) = \log p(s_{k^*(t)} \mid \mathcal{N}(\mu_{\text{in}}, \sigma_{\text{in}}^2)) - \log p(s_{k^*(t)} \mid \mathcal{N}(\mu_{\text{out}}, \sigma_{\text{out}}^2)), \tag{2}$$

where $s_{k^*(t)}$ is the score of the most informative public sample identified via Equation (1).

**Custom Voting (Image-based).** Image-based curation's deterministic nearest-neighbor structure $(s(x) = \max_t \cos(\phi(x), \phi(t)))$ enables reverse-engineering: for each public sample, identify which target $t^* = \arg\min_t |s(x) - \text{sim}(\phi(x), \phi(t))|$ was responsible, increment $v_{\text{vote}}(t^*)$ (positive evidence), and decrement votes for all $t$ where $\text{sim}(\phi(x), \phi(t)) > s(x)$ (negative evidence, *i.e.,* if $t$ had been present, $s(x)$ would be higher). The membership inference scores are the number of votes $v_{\text{vote}}$.

**Least Squares (TRAK).** TRAK-based curation computes scores as $s(x) = \frac{1}{|\mathcal{T}|} \sum_t \Phi(x)^\top G_t$. Because the score is a linear combination of per-target contributions, we can directly solve for the unknown membership mask that best explains the observed scores. We denote $m \in \{0, 1/|\mathcal{T}|\}^{|\mathcal{T}|}$ the masked mean operator. To recover membership signals, we then solve

$$\underset{m \in \mathbb{R}^n}{\text{minimize}} \quad \|\Phi(x)^\top G_t m - s\|_2^2. \tag{3}$$

The membership scores are the optimal weights $v_{\text{lstsq}} = m$. App. B.4.2 further details this.

## 3.4 Subset Selection Attacks

Assessing the privacy risks of the curated public subset is significantly more challenging than attacking the curation scores. This is because the scores yield a fine-grained ranking of the public samples whereas the curated dataset itself can be considered as a binary mask that only indicates for each public data point in the pool $\mathcal{D}$ whether it was included into the curated set $\tilde{\mathcal{D}}$.

**LiRA for Subsets.** While the original LiRA is designed to operate on continuous output logits, we need our attack to operate on *binary* selection observations. Using the shadow sets from Section 3.2, we binarize the curation outputs to top-$k$ masks $\{\overline{s}_i \in \{0, 1\}^N\}_{j=1}^m$ representing which public samples were selected.

In a naïve setup, we directly model these binary outcomes—whether each public sample was selected (1) or not (0) in each shadow curation—using Bernoulli distributions. For each $t \in \mathcal{T}$ we compute $\mu_{in} \in \mathbb{R}^N$ as the average of $s_i$ for every $t \in \mathcal{T}_i$ and $\mu_{out} \in \mathbb{R}^N$ where $t \notin \mathcal{T}_i$. These are the frequencies of public samples being in the top set depending on whether $t$ was part of the curation target. Using the $k^*$ filtering from Section 3.2, we select for each target $t$ the pool sample $x_t$ that is most indicative of $t$'s presence, *i.e.,* shows the highest difference between $\mu_{\text{in},t}$ and $\mu_{\text{out},t}$. We compute the membership inference score looking only the binary signal of whether $x_t$ is present or not as the log-likelihood ratio for the Bernoulli distribution

$$v_{\text{Binary LiRA}}(t) \log\left(\frac{P(x_v \mid \mu_{\text{in},t})}{P(x_v \mid \mu_{\text{out},t})}\right) = \log\left(\frac{\mu_{\text{in},t}^{x_v}(1 - \mu_{\text{in},t})^{1-x_v}}{\mu_{\text{out},t}^{x_v}(1 - \mu_{\text{out},t})^{1-x_v}}\right). \tag{4}$$

We also experiment with fitting the Bernoulli distribution with continuous labels (App. B.2).

**Voting-based Membership Inference (Image-based).** As an alternative to LiRA for Image-based curation, we exploit its deterministic nearest-neighbor structure. We define *overweighted* samples as those selected by our hypothesis but not by the target ($x \in \tilde{\mathcal{D}}_i$ but $x \notin \tilde{\mathcal{D}}$), suggesting our hypothesis includes incorrect targets. Conversely, *underweighted* samples ($x \in \tilde{\mathcal{D}}$ but $x \notin \tilde{\mathcal{D}}_i$) suggest missing targets in our hypothesis. We initialize $\tilde{\mathcal{T}}_0 = \{\emptyset\}$ and a vote accumulator $v_{\text{Iterative}}(t) = 0$ for each $t \in \mathcal{T}$. At iteration $i$:

$$\tilde{\mathcal{D}}_i = \text{Curate}(\mathcal{D}, \tilde{\mathcal{T}}_i), \quad \mathcal{O}_i = \tilde{\mathcal{D}}_i \setminus \tilde{\mathcal{D}}, \quad \mathcal{U}_i = \tilde{\mathcal{D}} \setminus \tilde{\mathcal{D}}_i \tag{5}$$

Update votes for each target $t \in \tilde{\mathcal{T}}_i$, where $t^*(x) = \arg\max_{t' \in \tilde{\mathcal{T}}_i} \text{sim}(\phi(x), \phi(t'))$ denotes the nearest target to $x$:

$$v_{\text{Iterative}}(t) \mathrel{+}= \sum_{x \in \mathcal{U}_i} \mathbb{1}[t = t^*(x)] - \sum_{x \in \mathcal{O}_i} \mathbb{1}[t = t^*(x)] \tag{6}$$

Keep targets with a positive sum of votes

$$\tilde{\mathcal{T}}_{i+1} = \{t \in \tilde{\mathcal{T}}_i : v_{\text{Iterative}}(t) \geq 0\}. \tag{7}$$

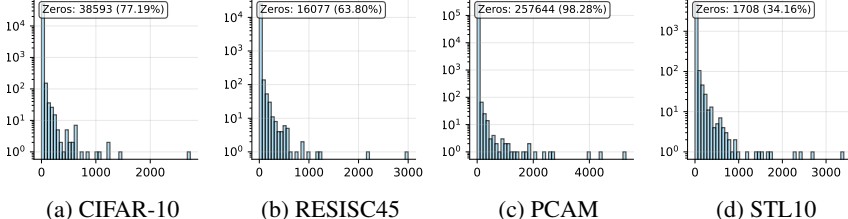

Figure 2: **Influence sparsity in Image-based curation.** Distribution of how many public samples have each target as their nearest neighbor. The concentration at zero demonstrates that most targets have no direct influence on curation scores, necessitating our fingerprinting approach.

The algorithm terminates when the curated set from our hypothesis closely matches the observed curated set ($J(\tilde{\mathcal{D}}_i, \tilde{\mathcal{D}}) \geq \theta$), where $J(\cdot, \cdot)$ denotes Jaccard similarity. We note that multiple target sets can produce identical curated outputs due to the many-to-one mapping from targets to selected samples. The final vote counts $v_{\text{Iterative}}(t)$ serve as membership inference scores: higher votes indicate stronger evidence that target $t$ was present in the private curation set.

For **TRAK-based curation**, we rely on the Binary LiRA scores $v_{\text{Binary LiRA}}$ from above, as TRAK's averaging does not expose the per-target structure that the voting attack exploits. We also evaluate a variant with continuous shadow labels (App. B.2).

## 3.5 FINAL MODELS ATTACK

Finally, we assess privacy leakage of the target data from the final model $\mathcal{M}$ trained purely on the curated public data. This represents the highest threat as such end-to-end attack could actually be instantiated against exposed models that are trained based on curated data.

To successfully extract membership information in this end-to-end setup, we rely on inserting a few modified samples $\mathcal{F}$ into the large curation pool. This is a realistic setup, as Carlini et al. (2024) have shown various practical ways of putting adversarial examples into web-scale training data. Samples in $\mathcal{F}$ must satisfy two requirements:

1. **Selective triggering:** Each sample must be selected during curation if and only if a specific target $t \in \mathcal{T}$ is present in the private target set. This requires crafting samples whose curation scores are sensitive to individual targets rather than the aggregate properties of $\mathcal{T}$.
2. **Detectable fingerprint:** Selected samples must imprint a measurable signal in the trained model $\mathcal{M}$ that the adversary can later detect to infer membership.

We empirically validate that fingerprinted samples imprint a detectable signal in the final model (Figures 10 and 11 in App. C.2): the signal does not degrade model utility but leads to elevated probabilities on semantically unrelated concepts. For as few as 5 fingerprint samples, the signal remains constant for training dataset sizes up to 1,000,000[1] (poisoning rate 0.0005%).

Since training large models for each experiment is intractable, we validate the signal once and reduce the end-to-end attack to observing whether the fingerprinted samples were selected during curation (*i.e.*, $f \in \tilde{\mathcal{D}}$). We now detail how we construct $\mathcal{F}$ and compute membership scores for each curation method.

In **Image-Based Curation**, selection from the pool depends solely on image embeddings. Hence, we can modify text captions arbitrarily without affecting curation scores. Being able to arbitrarily modify captions has been shown sufficient for imprinting a measurable signal in trained models (Carlini & Terzis, 2022). However, achieving selective triggering remains challenging due to the sparse influence structure of nearest-neighbor selection. As illustrated in Figure 2, in the unmodified setup, most target samples have zero influence on public sample scores under nearest-neighbor curation. *E.g.*, for PCAM and STL10, 98.28% and 34.16% of target samples are not the nearest neighbor for any public sample, respectively. This influence sparsity creates both a challenge (most targets are unattackable by default) and an opportunity (influenced samples have a strong membership signal).

---

[1]This is the maximum we evaluated, not necessarily where the signal breaks down.

---

**Algorithm 1** TRAK Membership Inference via Fingerprint Detection

---

**Require:** Pool data $X$, target gradients $Y$, fingerprints $\mathcal{F}$, threshold assumption $\rho$
**Ensure:** Membership scores for each target
$\quad G_\lambda^{-1} \leftarrow (X^\top X + \lambda I)^{-1}$ {Cholesky decomposition}
$\quad \mu \leftarrow \mathbb{E}[Y], \Sigma \leftarrow \mathrm{Cov}[Y]$ {Target statistics}
$\quad S \leftarrow \mathcal{F} G_\lambda^{-1} Y^\top$ {Signal matrix}
$\quad \nu_i \leftarrow \sqrt{f_i^\top G_\lambda^{-1} \Sigma G_\lambda^{-1} f_i}$ for all $i$ {Noise scales}
$\quad$ Initialize $\mathrm{score}_j \leftarrow 0$ for all targets
$\quad$ **for** each target $j \in [n]$ **do**
$\quad\quad i^*(j) \leftarrow \arg\max_i |S_{ij}|/\nu_i$ {Best fingerprint}
$\quad\quad$ Compute $z_{H_0}, z_{H_1}$ via Sherman-Morrison
$\quad\quad p_{H_0}, p_{H_1} \leftarrow$ percentile ranks of $z_{H_0}, z_{H_1}$
$\quad\quad \mathrm{confidence}_j \leftarrow |\mathrm{clip}(p_{H_1}, \rho, 100) - \mathrm{clip}(p_{H_0}, \rho, 100)|$
$\quad$ **end for**
$\quad$ **for** each observed fingerprint $f_i \in \tilde{\mathcal{D}}$ **do**
$\quad\quad$ **for** each target $j$ where $i^*(j) = i$ **do**
$\quad\quad\quad$ **if** $p_{H_1} > \rho \land p_{H_0} \leq \rho$ **then** {Fingerprint crosses threshold}
$\quad\quad\quad\quad \mathrm{score}_j \leftarrow \mathrm{score}_j + \mathrm{confidence}_j$
$\quad\quad\quad$ **end if**
$\quad\quad$ **end for**
$\quad$ **end for**
$\quad$ **return** $\{\mathrm{score}_j\}_{j=1}^n$

---

We construct candidates for $\mathcal{F}$ from images in $\mathcal{D}$ paired with semantically unrelated captions (*e.g.*, "ratatouille" for CIFAR-10 targets). Unrelated captions ensure detectability: if selected, these fingerprints imprint a measurable signal in the trained model that would not appear otherwise. For each candidate fingerprint $f$ and target $t_i$, we compute a correspondence score balancing two objectives. Let $s(a, b) = \mathrm{sim}(\phi(a), \phi(b))$ denote embedding similarity, then

$$\mathrm{score}(f, t_i) = \alpha \cdot \underbrace{s(f, t_i)}_{\text{attraction to } t_i} + (1-\alpha) \cdot \underbrace{\left(1 - \max_{t' \neq t_i} s(f, t')\right)}_{\text{repulsion from } t' \neq t_i} \tag{8}$$

where $\alpha \in [0, 1]$ trades off between proximity to the intended target and separation from samples in the target set. Our attack inserts as fingerprints $\mathcal{F} = \{\arg\max_{f \in \mathcal{D}} \mathrm{score}(f, t_i) | t_i \in \mathcal{T}\}$, *i.e.*, the highest-scoring sample for each target (not always uniquely, *i.e.*, $|\mathcal{F}| \leq |\mathcal{T}|$).

Given the observed fingerprint selections, we compute membership scores as follows: Let $p_0^f$ denote the baseline percentile rank of fingerprinted sample $f$ when scored against the full target set $\mathcal{T}$. The adversary does not know the selection threshold $\tau$ but can conservatively assume they will pick from the top $50\%$ of the pool. We model the probability of selecting $f$ without its corresponding target as:

$$P_0(f) = \frac{1}{1 + \exp(-(p_0^f - \tau)/\sigma)}, \tag{9}$$

where $\sigma$ controls the transition sharpness. For each fingerprinted sample we measure to be in $\tilde{\mathcal{D}}$ we set the membership inference score of the corresponding target(s)[2] to the *surprise* of the measurement of that sample under $H_0$ divided by the number of targets that share that fingerprint.

$$v_{\text{E2E Img.}}(t_i) = \frac{1[\text{ selected }] - P(\text{ selected } | P_0(f))}{n_{\text{sharing}}(f)}. \tag{10}$$

For **TRAK-based Curation**, we face the challenge that TRAK explicitly penalizes mislabeled samples through gradient alignment scoring (Park et al., 2023), rendering direct caption manipulation ineffective. However, we discover that appending semantically orthogonal information to correct

---

[2]Target(s) in plural, as the mapping is not necessarily unique.

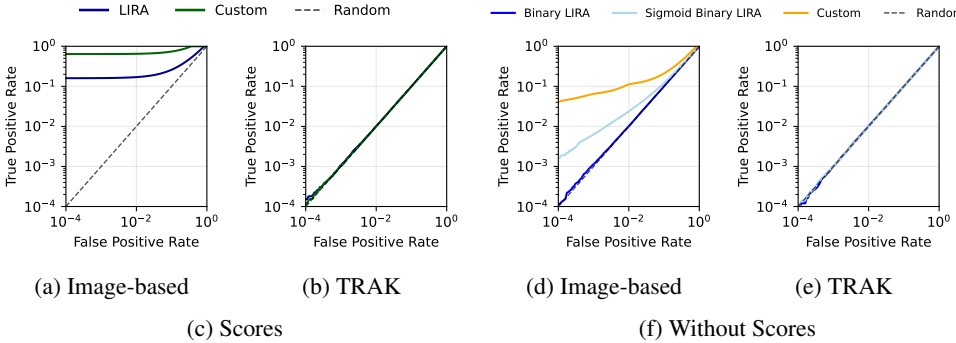

Figure 3: **Attack success for curation scores and subsets.** Image-based curation's nearest-neighbor mechanism is highly vulnerable, while TRAK's gradient averaging shows almost no leakage.

captions (*e.g.,* "an image of an airplane and ratatouille") preserves high TRAK scores while enabling detectable model changes (Figure 11). This works because TRAK evaluates gradient alignment along task-relevant dimensions; orthogonal additions contribute negligible projection onto these directions.

We construct fingerprint candidates by copying target samples and augmenting their captions with orthogonal signals. For each candidate-target pair, we compute a signal-to-noise ratio measuring how sensitively the fingerprint's score responds to that target's presence versus interference from other targets. We select the highest-SNR fingerprint for each target to form $\mathcal{F}$. The membership score for target $t_j$ depends on whether its fingerprint crosses the selection threshold when $t_j$ is present but not when absent:

$$v_{\text{E2E Trak}}(t_j) = \begin{cases} \text{confidence}_j & \text{if } f_{i^*(j)} \in \tilde{\mathcal{D}} \\ 0 & \text{otherwise} \end{cases}, \tag{11}$$

where $\text{confidence}_j$ quantifies the percentile shift between the two hypotheses. Algorithm 1 provides the complete TRAK attack procedure; full mathematical details appear in App. C.3. The Image-based attack procedures are detailed in Algorithms 4 and 5.

### 3.6 SUMMARY OF ATTACKS

We developed **7 attacks** across the three pipeline stages. Our **Score-Based Attacks (Section 3.3)** infer membership information from the curation scores. The **(1) LiRA**-based attack adapts likelihood ratio tests by running shadow curation processes and comparing score distributions between member and non-member scenarios for both Image-based and TRAK-based methods. **(2) Voting Scheme** exploits Image-based curation by reverse-engineering nearest-neighbor relationships through voting across shadow predictions. **(3) Least Squares** solves linear systems that relate TRAK-based curation scores to target membership. Our **Subset Selection Attacks (Section 3.4)** target the binary selection decisions made by curation. **(4) Binary LiRA** adapts the likelihood ratio framework to binary subset selection by modeling selection probabilities as Bernoulli distributions and comparing likelihoods under member versus non-member hypotheses. **(5) Iterative Voting Scheme** iteratively refines a hypothesis of the target set by running curation on candidate sets and adjusting the hypothesis until the resulting selection matches that of the target set. Finally, our **End-to-End Model Attacks (Section 3.5)** exploit the downstream model trained on curated data. **(6) Fingerprinting (Image-based)** uses mislabeled captions as fingerprints, selecting them via correspondence scoring to ensure they appear only when specific private examples are present in curation. **(7) Fingerprinting (TRAK)** employs benign captions with added orthogonal information as fingerprints, selecting them based on membership signal-to-noise ratio and selection chance.

## 4 EVALUATION

**Experimental Setup.** We evaluate our attacks using CommonPool (small) (Gadre et al., 2023) as the public dataset $\mathcal{D}$, containing 12.8M samples. We evaluate across six diverse target datasets spanning natural images, medical imaging, and satellite imagery to cover a wide range of curation setups:

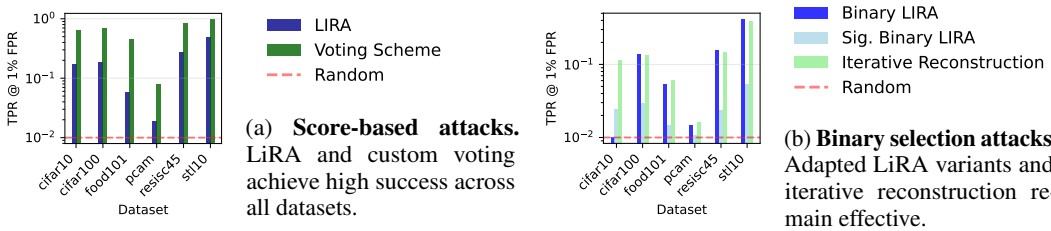

Figure 4: **Attack success correlates with influence patterns from Figure 2.** Cross-dataset comparison shows TPR at 1% FPR inversely correlates with the percentage of zero-influence targets.

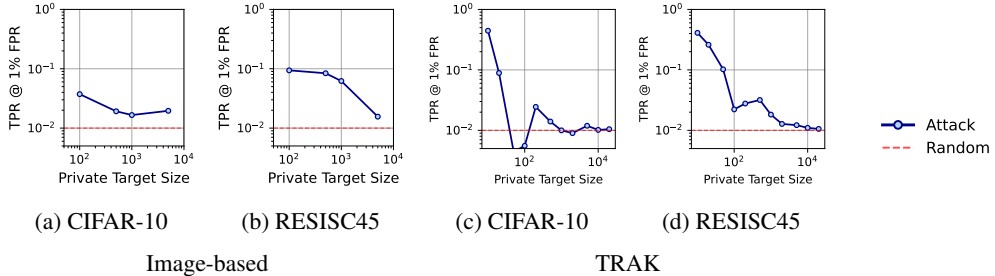

Figure 5: **End-to-end membership inference success.** Image-based curation shows consistent partial leakage, while TRAK exhibits size-dependent vulnerability.

CIFAR-10/100 (Krizhevsky, 2009), STL-10 (Coates et al., 2011), RESISC45 (Cheng et al., 2017), PatchCamelyon (Veeling et al., 2018), and Food101 (Bossard et al., 2014). For end-to-end attacks, we train models following *DataComp* small-scale (see Section C for full details). We obtain image embeddings from OpenAI's CLIP ViT-L/14 model (Radford et al., 2021). Gradients for TRAK are obtained from a model trained on the full pool dataset $\mathcal{D}$ using contrastive training as in Gadre et al. (2023). For LIRA attacks, we use 256 shadow sets per configuration (see Section C for details).

### 4.1 ATTACK PERFORMANCE ON CURATION SCORES

Image-based curation exhibits high attack success rates. The dataset-specific variations in attack success for Image-based curation align with our analysis from Figure 2, namely that datasets with higher influence concentration (fewer targets affecting many public samples) exhibit greater vulnerability. This holds regardless the underlying data type—satellite imagery (RESISC45) with less zero-influence samples is more attackable and medical images (PatchCamelyon) with more zero-influence samples is less attackable than CIFAR10. In contrast, TRAK demonstrates natural protection through its averaging mechanism, giving near-random attack performance (AUC $\approx 0.5$) as individual target contributions become diluted through aggregation and dimensionality reduction.

### 4.2 ATTACK PERFORMANCE ON BINARY SELECTIONS

Even when restricted to observing only binary selection patterns, Image-based curation remains vulnerable to membership inference attacks (Section 4). Our iterative reconstruction algorithm successfully recovers the private target set for all samples with non-zero influence, though the large fraction of zero-influence samples provides natural protection for the remaining targets. This success generalizes over various datasets (Figure 4b).

### 4.3 END-TO-END ATTACKS ON TRAINED MODELS

The end-to-end attack depends significantly on the target dataset size, so we evaluate TPR at $1\%$ FPR for different target dataset sizes $|\mathcal{T}|$ and various datasets in Figure 5. We show that Image-based curation leads to moderate information leakage across all target dataset sizes, with TPR values up to 21.4% at 1% FPR for RESISC45 at $|\mathcal{T}| = 100$. Targets with non-zero influence are exposed at every target size, creating a bimodal privacy distribution where most samples remain protected while

a subset faces significant exposure. TRAK exhibits a different behavior, demonstrating a strong decrease in attack success with growing $|\mathcal{T}|$. This size-dependent vulnerability suggests TRAK may be suitable for large-scale applications but poses significant risks for small, sensitive datasets.

## 4.4 ABLATIONS

We perform further ablations on various aspects of our curation setups and analyze their impact on the attack success. We conduct these ablations on attacks against scores, which yield the strongest leakage signal and thus provide the clearest insights into the factors affecting attack success. First, we analyze the impact of **the number of dimensions** used for Image-based embeddings and TRAK-based gradient projections. Our results in Figure 37 suggest that for Image-based curation, there is a sweet spot, with the highest leakage at 128 dimensions. TRAK requires enough dimensions ($\geq 1{,}024$), below which the attack success drops. Additionally, we show how **target dataset size** impacts attack success. Our results in Figure 32 suggest differences across datasets: *e.g.,* attack success against STL-10 remains near constant while for CIFAR-10, we observe that the success drops as the target

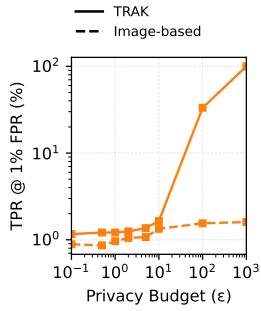

Figure 6: Attack success versus $\varepsilon$ for CIFAR-10 ($|\mathcal{T}| = 1000$).

dataset size grows. Finally, Figure 36 shows that the **number of shadow models for LiRA** has a varying effect on improving the attack success on different datasets for Image-based curation: *e.g.,* double the number of shadow models from 128 to 256 for CIFAR-10 increases AUC by (abs.) 10% while for PCAM only 1%. For TRAK, we do not observe any improvement in attack success when adding more shadow models. In App. E, we provide: (1) full ROC curves across all six datasets for score-based (Figures 14 and 16), selection-based (Figures 15 and 17) and end-to-end attacks (Figures 18 and 19 and Figures 26 to 31), (2) target size ablations showing how larger target datasets provide a shielding effect (Figures 32 and 34), (3) shadow model scaling showing different improvements for different datasets (Figure 36), (4) embedding dimension ablations revealing optimal attack configurations (Figure 37), and (5) caption design ablations for TRAK fingerprints (App. C.5).

## 5 MITIGATING PRIVACY LEAKAGE

Having demonstrated that data curation pipelines pose real privacy risks (Section 4), we now turn to mitigations. Differential privacy (DP) is the de facto standard for providing formal privacy guarantees (Dwork et al., 2006a). We propose DP adaptations of both curation methods to yield $(\varepsilon, \delta)$-DP guarantees via the Gaussian mechanism (Dwork et al., 2006b). For **DP Image-based curation** we add calibrated Gaussian noise to each per-target similarity before selecting the maximum (Report Noisy Max (Dwork et al., 2014)). For **DP TRAK-based curation** we privatize the mean gradient computation, similar to Abadi et al. (2016). App. D.1 details the algorithms. Our DP adaptations mitigate leakage (App. 4): at $\varepsilon = 100$, TPR at 1% FPR drops from 98.4% to 5.4% for Image-based and from 100% to 33.2% for TRAK, which requires stricter guarantees due to its higher-dimensional gradient space; at $\varepsilon = 10$, both drop to near-baseline (1.1% and 1.7%, respectively).

We further show in App. D.2 that simply **removing the most vulnerable samples** does not prevent leakage, exhibiting a *privacy onion effect* (Carlini et al., 2022b) for Image-based curation.

## 6 DISCUSSION AND CONCLUSION

Our work demonstrates that data curation pipelines can leak membership information about the target datasets, exposing privacy risks at every stage, from curation scores and curated subset to the final trained model. Image-based nearest-neighbor methods are particularly vulnerable, and even state-of-the-art approaches like TRAK expose privacy risks for small target sets. Our discovered risks become practically relevant when adversaries can introduce manipulated samples in the curation pool, which is the case when the pool is simply crawled from the internet. This highlights that, as curation becomes central to ML, we need novel curation methods with dedicated safeguards, such as Differential Privacy (DP), to reduce potential leakage throughout the entire data curation pipeline.

## ACKNOWLEDGEMENTS

We would like to acknowledge our sponsors, who support our research with financial and in-kind contributions: OpenAI and G-Research. We also thank members of the SprintML[3] group for their feedback. Responsibility for the content of this publication lies with the authors. This work was supported by the Helmholtz Association's Initiative and Networking Fund on the HAICORE@FZJ partition.

## ETHICS STATEMENT

Our work demonstrates privacy vulnerabilities in data curation methods. We conduct this research to identify and quantify these issues before they can be exploited maliciously, enabling the development of privacy-preserving methods. Our attacks require specific capabilities like injecting target-specific and modified samples into public datasets, which limit their immediate applicability and provide time to implement countermeasures.

## REPRODUCIBILITY STATEMENT

Reproducing our results is possible from the information provided, and we support reproduction through the release of all our attack implementations in the supplemental code, as well as providing more detail on the algorithms in the appendix. What hinders reproduction is the computational cost associated with running curation and training algorithms on dataset sizes that are relevant to the topic of data curation. We argue that this is a) to some extent unavoidable and b) our introduction of suitable proxy metrics that eliminate the need to train models improves reproducibility significantly.

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

# A CURATION METHODS

Table 2: Overview of Curation Methods

| Method | Description | Score Function |
|---|---|---|
| Image-based | Identifies relevant pool data using image embeddings | $s(x) = \max_{t \in \mathcal{T}} \cos(\phi(x), \phi(t))$ where $\phi(\cdot)$ is the embedding function |
| TRAK | Computes attribution scores via projected gradients to identify influential pool samples | $s(x) = \frac{1}{|\mathcal{T}|} \sum_{t \in \mathcal{T}} \Phi(x)^T G_t$ where $\Phi$ are projected and out-to-loss-scaled features |

## A.1 IMAGE-BASED SCORING

We employ Image-based nearest neighbor distances inspired by Gadre et al. (2023). Following the DataComp methodology, we use embeddings from OpenAI's pretrained CLIP ViT-L/14 model (Radford et al., 2021) to compute the image representations $\phi(\cdot)$. For each pool sample $x_i \in \mathcal{D}$, the curation score is computed as the maximum cosine similarity to any target sample in $\mathcal{T}$, *i.e.,* the pool sample is scored based on its closest (most similar) target sample. This nearest-neighbor mechanism creates a deterministic relationship where each pool sample's score is determined by exactly one target sample.

## A.2 TRAK

The TRAK algorithm (Park et al., 2023) for computing influence scores is formally presented in Algorithm 2.

---

**Algorithm 2** TRAK Algorithm for Influence Computation

---

**Require:** Training dataset $\mathcal{D}$, target dataset $\mathcal{T}$, model $f_\theta$, projection dimension $d_{\text{proj}}$
**Ensure:** Influence scores $s$ of training samples on target samples
$\quad \mathbf{G}_{\text{train}} \leftarrow \nabla_\theta f_\theta(\mathcal{D})$ {Gradients for training data}
$\quad \boldsymbol{\alpha}_{\text{train}} \leftarrow \nabla_{f_\theta} \mathcal{L}(\mathcal{D}, f_\theta)$ {Output-to-loss gradients}
$\quad \mathbf{G}_{\text{target}} \leftarrow \nabla_\theta f_\theta(\mathcal{T})$ {Gradients for target data}
$\quad \mathbf{P} \leftarrow \text{RandomProjectionMatrix}(\dim(\theta), d_{\text{proj}})$ {Typically Rademacher}
$\quad \mathbf{G}_{\text{train}} \leftarrow \mathbf{G}_{\text{train}} \mathbf{P}$ {Project training gradients}
$\quad \mathbf{G}_{\text{target}} \leftarrow \mathbf{G}_{\text{target}} \mathbf{P}$ {Project target gradients}
$\quad \mathbf{X}^T \mathbf{X} \leftarrow \mathbf{G}_{\text{train}}^T \mathbf{G}_{\text{train}}$ {Compute Gram matrix}
$\quad \boldsymbol{\Phi} \leftarrow \mathbf{G}_{\text{train}} \cdot (\mathbf{X}^T \mathbf{X})^{-1}$ {Compute features}
$\quad \mathbf{s}_{\text{raw}} \leftarrow \frac{1}{|\mathcal{T}|} \sum_{i=1}^{|\mathcal{T}|} \boldsymbol{\Phi} \cdot \mathbf{G}_{\text{target},i}^T$ {Raw scores}
$\quad \mathbf{s} \leftarrow \mathbf{s}_{\text{raw}} \odot \boldsymbol{\alpha}_{\text{train}}$ {Scale by output-to-loss gradients}
$\quad \textbf{return } \mathbf{s}$ {Final influence scores}

---

# B ATTACKING CURATION SCORES

## B.1 ATTACK DESIGN RATIONALE

We design our attacks to exploit the specific mathematical structure of each curation method. While no attack can be proven universally optimal without strong distributional assumptions, our approaches are theoretically grounded and, in the case of Image-based scoring, exploit fundamental limits of what can be inferred from the exposed information.

**Method-Agnostic Attacks via LiRA.** For general-purpose membership inference, we employ the Likelihood Ratio Attack (LiRA) (Carlini et al., 2022a). This attack is theoretically motivated by the

Neyman-Pearson lemma, which establishes that the likelihood ratio test is optimal for binary hypothesis testing (membership vs. non-membership) under the assumption that the score distributions are known or can be accurately estimated. LiRA provides a principled baseline that makes minimal assumptions about the curation method's internal structure.

We make two key adaptations for curation pipelines. First, we replace shadow *models* with shadow *curation runs,i.e.,* instantiations of the curation algorithm using different random subsets of $\mathcal{T}$, s.t. each target sample is part of exactly half of the random subsets. Second, following Jagielski et al. (2023), we filter the curation scores to select only the public sample per target with the maximum difference in expected scores between member and non-member hypotheses (Equation (1)).

This preserves the essential statistical properties needed for LiRA: we can empirically model $P(s|t \in \mathcal{T})$ and $P(s|t \notin \mathcal{T})$ from shadow observations. Figure 7a illustrates the resulting in and out score distributions. We can then formulate a likelihood ratio attack on the curation scores by computing the log-ratio of the in and out score distributions (Equation (2)). Figure 7b illustrates the mapping from measured scores to the resulting membership inference score. Given sufficient shadow sets and correct distributional assumptions (*e.g.,* Gaussian for continuous scores, Bernoulli for binary selections), LiRA approximates the theoretically optimal likelihood ratio test.

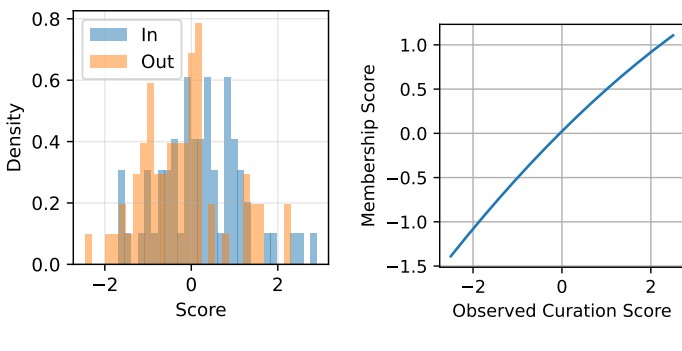

(a) In and out score distributions.  (b) Membership inference score.

Figure 7: **LiRA can be adapted to curation pipelines.** (a) We can empirically model $P(s|t \in \mathcal{T})$ and $P(s|t \notin \mathcal{T})$ from shadow observations. (b) We can then compute membership inference scores based on a likelihood ratio attack by computing the log-ratio of the in and out score distributions.

**Image-Based Curation: Exploiting Deterministic Structure.** For Image-based scoring with nearest-neighbor retrieval ($s(x) = \max_{t \in \mathcal{T}} \cos(\phi(x), \phi(t))$), the max operation creates a *deterministic* function where each pool sample's score is determined by exactly one target sample: its nearest neighbor. This structure enables perfect reverse-engineering: given $s(x)$ and the embeddings, we can identify which $t^* \in \mathcal{T}$ was responsible via $t^* = \arg\min_t |\cos(\phi(x), \phi(t)) - s(x)|$.

Our custom voting attack exploits this theoretical limit, since only nearest-neighbor relationships are exposed, and our attack recovers exactly these relationships. Positive votes accumulate for targets that are nearest neighbors to pool samples, while negative votes identify targets that would have produced higher scores if present. This is fundamentally more powerful than LiRA because it does not rely on distributional assumptions or shadow set approximations; it deterministically extracts the membership signal embedded in the nearest-neighbor structure.

The attack's effectiveness is constrained only by: (1) the sparsity of nearest-neighbor relationships (many targets may not be nearest neighbors to any pool sample, as shown in the main paper), and (2) numerical precision in matching scores to similarities. Unlike LiRA, which requires many shadow sets to approximate distributions, our deterministic attack needs only the scoring function's mathematical structure and embedding access.

### B.1.1 ORACLE ATTACK SETUP

To establish an upper bound on attack performance for Image-based scoring, we define an oracle attack that leverages perfect knowledge of the curation mechanism. The oracle operates under the following membership scoring scheme for each target sample $t \in \mathcal{T}$:

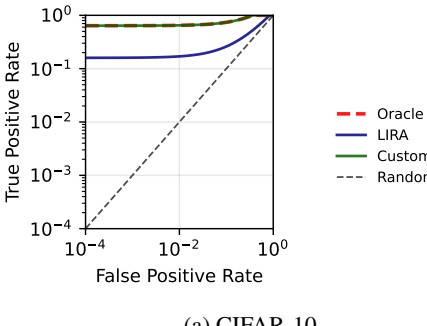

(a) CIFAR-10

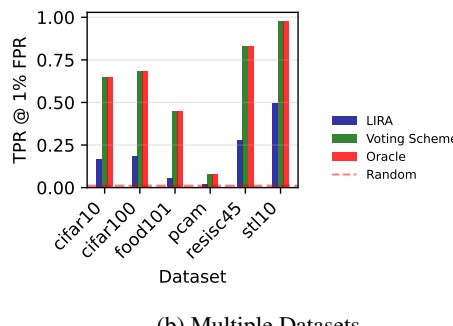

(b) Multiple Datasets

Figure 8: **Our custom attack on Image-based scores matches oracle performance.** With access to the scores, the attack achieves the same utility as the oracle attack.

$$
\text{membership\_score}(t) = \begin{cases} 1 & \text{if } t \text{ is the nearest neighbor of any pool sample} \\ 0 & \text{if } \exists x \in \mathcal{D} : \cos(\phi(x), \phi(t)) > s(x) \\ 0.5 & \text{otherwise (no influence on scores)} \end{cases} \tag{12}
$$

The rationale for this scoring scheme is as follows:

- **Score 1 (Member):** If a target sample $t$ is responsible for the observed score of at least one pool sample (*i.e.,* it is the nearest neighbor in $\mathcal{T}$ for that pool sample), then the oracle has definitive evidence that $t \in \mathcal{T}_{\text{sel}}$.

- **Score 0 (Non-member):** If the oracle can identify pool samples where the similarity to $t$ exceeds the observed score, this indicates that $t$ would have produced a higher score if it were in $\mathcal{T}_{\text{sel}}$. Since it did not, the oracle can definitively conclude that $t \notin \mathcal{T}_{\text{sel}}$.

- **Score 0.5 (Unknown):** For target samples that have no observable influence on any pool scores—neither as nearest neighbors nor as potential higher-scoring alternatives—even the most powerful oracle can only guess their membership status uniformly at random.

This oracle attack represents the theoretical maximum achievable performance given the information exposed by the Image-based scoring mechanism. As shown in Figure 8, our deterministic voting attack matches oracle performance when scores are available, confirming that it successfully extracts all membership information embedded in the nearest-neighbor structure.

**TRAK: Compressed Sensing for Averaged Signals.** TRAK scores are computed as $s(x) = \frac{1}{|\mathcal{T}|} \sum_{t \in \mathcal{T}} \Phi(x)^{\top} G_t$, averaging gradient contributions projected to a low-dimensional space from all targets. This averaging *diffuses* individual membership signals, preventing the deterministic reverse-engineering that succeeds for Image-based methods.

Recovering which targets contributed to averaged scores is a *sparse recovery* problem: given the observation $s \in \mathbb{R}^{|\mathcal{D}|}$ and the measurement matrix $\Phi^{\top} G \in \mathbb{R}^{|\mathcal{D}| \times |\mathcal{T}|}$, we seek the sparse indicator vector $\mathcal{T}_{\text{sel}} \in \{0, 1\}^{|\mathcal{T}|}$ such that $s \approx \Phi^{\top} G \mathcal{T}_{\text{sel}}$. This is a compressed sensing problem that has been extensively studied in signal processing and optimization literature.

We employ Orthogonal Matching Pursuit (OMP) and Iterative Hard Thresholding (IHT) (Axiotis & Sviridenko, 2022)—greedy algorithms known to provide near-optimal solutions under certain conditions. These represent the state-of-the-art for practical sparse recovery. Theoretically optimal attacks would require solving the $\ell_0$-minimization problem exactly, which is NP-hard (Natarajan, 1995). Brute-force search over all $\binom{|\mathcal{T}|}{|\mathcal{T}_{\text{sel}}|}$ possible target subsets is computationally infeasible for realistic dataset sizes (*e.g.,* $\binom{50000}{5000} \approx 10^{7000}$ possibilities).

In practice, we find that even these theoretically-grounded compressed sensing attacks fail to meaningfully outperform LiRA on TRAK. This empirical result suggests that the averaging operation in TRAK provides inherent protection by spreading signal across many dimensions, making sparse recovery ill-conditioned. The attack success becomes dependent on $|\mathcal{T}|$: smaller target sets (common

in sensitive domains) concentrate signal and become more vulnerable, while larger sets benefit from stronger averaging.

## B.2 Method-Agnostic Attacks

To attack the curation methods, we employ the likelihood-ratio attack (LiRA) from Carlini et al. (2022a). We initialize $N = 256$ shadow models. Each of those uniformly curates for a random subset of the target dataset, s.t., each target sample is used in exactly half of the shadow models. Furthermore, we perform 25 independent curation runs on random subsets that we will attack.

To analyze the attacakbility of a target sample, we group the shadow model scores into those that contained the target sample $\mathbf{S}_{\text{in}}$ and those that did not $\mathbf{S}_{\text{out}}$. We then average over the shadow models, giving us two vectors of pool scores $\mathbf{s}_{\text{in}}$ and $\mathbf{s}_{\text{out}}$. As many of those scores will probably not give us meaningful membership signals (Jagielski et al., 2023), we first find the index of the pool score that has the largest average difference, *i.e.,* $i = \arg\max |\mathbf{s}_{\text{in}} - \mathbf{s}_{\text{out}}|$. We then fit a Gaussian to the distribution of $\mathbf{S}_{i,\text{in}}$ and $\mathbf{S}_{i,\text{out}}$, giving us two distributions $P_{\text{in}}(s)$ and $P_{\text{out}}(s)$. We can then compute the log-ratio of the of the $i$th pool score to the two distributions, giving us the LiRA score.

To perform the attack without access to the scores, we replace the scores with binary signals, indicating whether a particular pool sample would be in the selected partition of the shadow models. The **binarized LiRA** computes the mean of those binary signals and computes the difference in bernoulli log-probability mass functions.

**Soft Binarization.** While the adversary only observes binary selections for the target, since they set up the shadow curations themselves, they have access to the underlying continuous scores for those. Rather than discarding this information, we apply sigmoid transformation $\tilde{s} = \sigma_{\pi_k}(s_i) = 1/(1 + \exp(-\gamma \cdot (s_i - \pi_k)/\tau))$ to preserve boundary proximity information. Empirically, we show that this improves attack success when samples cluster near the selection threshold but provides no benefit when scores are clearly separated (Figure 35).

## B.3 Image-Based Scoring

We recall that for Image-based scoring, the score for each sample in $\mathcal{D}$ is only influenced by the nearest neighbour in $\mathcal{T}$. Therefore, we analyse how many pool samples each target sample is the nearest neighbour for.

Given a pool sample $x \in \mathcal{D}$ with target score $s(x)$, we can determine which target sample $t \in \mathcal{T}$ was responsible for this score through the following attack:

1. Normalize the embeddings: $\hat{x} = \frac{\phi(x)}{\|\phi(x)\|_2}$ and $\hat{t} = \frac{\phi(t)}{\|\phi(t)\|_2}$ for all $t \in \mathcal{T}$

2. Compute similarities: $\text{sim}(x,t) = \hat{x}^T \hat{t}$ for all $t \in \mathcal{T}$

3. Find target with matching score: $t^* = \arg\min_{t \in \mathcal{T}} |\text{sim}(x,t) - s(x)|$

4. For each target $t$, assign votes:

$$v(t) = \begin{cases} 1 & \text{if } t = t^* \\ -1 & \text{if } \text{sim}(x,t) > s(x) \\ 0 & \text{otherwise} \end{cases}$$

The voting scheme reveals membership in $\mathcal{T}$ through positive votes: if a target sample receives positive votes, it was used to compute the score of at least one pool sample. Negative votes indicate that this target sample could have given a higher score than what was observed, implying it was not used in the curation.

**Selection-Based Attack.** When we only have access to the selected pool samples $\tilde{\mathcal{D}} \subseteq \mathcal{D}$ but not their scores, we can still reconstruct the target dataset through an iterative elimination process:

1. Initialize $\mathcal{T}_0 = \mathcal{T}$ as the full target dataset, $\mathcal{R} = \emptyset$ as the set of removed samples, and voting scores $\mathbf{v} = \mathbf{0}$

---

**Algorithm 3** Image-based Scores MIA

---

**Input:** Dataset embeddings $\{\phi(x)\}_{x \in \mathcal{D}}$, target embeddings $\{\phi(t)\}_{t \in \mathcal{T}}$, target scores $\mathbf{s}$
**Output:** Votes $v(t)$ for each target $t \in \mathcal{T}$
Initialize $\forall_{t \in \mathcal{T}} : v(t) = 0$

Normalize all target embeddings: $\hat{\mathbf{t}} = \frac{\phi(t)}{\|\phi(t)\|_2}$ for all $t \in \mathcal{T}$

**for** each sample $x \in \mathcal{D}$ **do**
  Normalize the query embedding: $\hat{\mathbf{x}} = \frac{\phi(x)}{\|\phi(x)\|_2}$
  Compute similarities: $\mathrm{sim}(x, t) = \hat{\mathbf{x}}^T \hat{\mathbf{t}}$ for all $t \in \mathcal{T}$
  Find target with matching score: $t^* = \arg\min_{t \in \mathcal{T}} |\mathrm{sim}(x, t) - s(x)|$
  **for** each target $t \in \mathcal{T}$ **do**
    **if** $t = t^*$ **then**
      $v(t) = v(t) + 1$
    **else if** $\mathrm{sim}(x, t) > s(x)$ **then**
      $v(t) = v(t) - 1$
    **end if**
  **end for**
**end for**

---

2. At each iteration $i$:
   (a) Identify target-missing and target-exclusive samples:
      - Target-missing: $\mathcal{M}_i = \tilde{\mathcal{D}}_i \setminus \tilde{\mathcal{D}}_{\text{target}}$ (selected by us but not target)
      - Target-exclusive: $\mathcal{E}_i = \tilde{\mathcal{D}}_{\text{target}} \setminus \tilde{\mathcal{D}}_i$ (selected by target but not us)
   (b) Find nearest neighbors in remaining targets:
      - For target-missing: $\mathcal{N}_i^- = \mathrm{NN}(\mathcal{M}_i, \mathcal{T}_i)$
      - For target-exclusive: $\mathcal{N}_i^+ = \mathrm{NN}(\mathcal{E}_i, \mathcal{T}_i)$
   (c) Update votes:
      - $\mathbf{v}[\mathcal{N}_i^-] \leftarrow \mathbf{v}[\mathcal{N}_i^-] - 1$ (negative for likely non-members)
      - $\mathbf{v}[\mathcal{N}_i^+] \leftarrow \mathbf{v}[\mathcal{N}_i^+] + 1$ (positive for likely members)
   (d) Update removed set: $\mathcal{R} = \{t \in \mathcal{T} : \mathbf{v}[t] < 0\}$
   (e) Update remaining targets: $\mathcal{T}_{i+1} = \mathcal{T} \setminus \mathcal{R}$
   (f) Recompute pool selection: $\tilde{\mathcal{D}}_{i+1} = \mathrm{TopK}(\mathrm{NN\text{-}Sim}(\mathcal{D}, \mathcal{T}_{i+1}), k)$
   (g) Compute Jaccard similarity: $J_i = \frac{|\tilde{\mathcal{D}}_{i+1} \cap \tilde{\mathcal{D}}_{\text{target}}|}{|\tilde{\mathcal{D}}_{i+1} \cup \tilde{\mathcal{D}}_{\text{target}}|}$

3. Stop when either:
   - Perfect reconstruction: $J_i = 1$
   - No improvement: $J_i - J_{i-p} < \epsilon$ for patience $p$ and threshold $\epsilon$

The attack maintains a voting score for each target sample. Negative votes suggest the target sample is causing overselection of pool samples that are no in $\mathcal{T}_{\text{Sel.}}$, while positive votes indicate the target sample helps select pool samples that are also in $\mathcal{T}_{\text{Sel.}}$. The ROC-AUC of these votes against true membership provides a measure of attack success.

## B.4 TRAK

TRAK computes the scores for all pool samples $\mathbf{s} \in \mathcal{R}^{|\mathcal{D}|}$ as

$$\mathbf{s} = \frac{1}{|\mathcal{T}|} \sum_{t \in \mathcal{T}} \mathbf{\Phi}^T \mathbf{G}_t \tag{13}$$

Following the DsDm methodology (Engstrom et al., 2024), we first train a CLIP model on the full pool dataset $\mathcal{D}$ to obtain a reference model. We then obtain the gradients $\mathbf{G}_t$ and $\mathbf{\Phi}$ from this model by computing the per-sample zero-shot gradients—the gradients obtained by minimizing the classification loss of the linear zero-shot classifier obtained by embedding caption-templates (Cherti

& Beaumont, 2025). The curation score for each pool sample is the average of its TRAK attribution scores across all target samples $t \in \mathcal{T}$, capturing how influential that pool sample would be for improving performance on the target dataset.

Since the target model does not use the entire target dataset, but instead an unknown subset $\mathcal{T}_{\text{sel}} \subseteq \mathcal{T}$, the target scores are computed as

$$\mathbf{s}_{\mathcal{T}_{\text{sel}}} = \frac{1}{|\mathcal{T}_{\text{sel}}|} \sum_{t \in \mathcal{T}_{\text{sel}}} \mathbf{\Phi}^T \mathbf{G}_t \tag{14}$$

where $\mathbf{G}_{\mathcal{T}_{\text{sel}}} \in \mathbb{R}^{d \times |\mathcal{T}_{\text{sel}}|}$ is the matrix of the target gradients for the selected target samples.

Recovering $\mathcal{T}_{\text{sel}}$ from $\mathbf{s}_{\mathcal{T}}$ constitutes a subset-sum problem, which is NP-hard (Natarajan, 1995). We know about $\mathcal{T}_{\text{sel}}$ that it is a subset of $\mathcal{T}$.

### B.4.1 Orthogonal Matching Pursuit

Hence, we can formulate this as a standard compressed sensing problem with the formulation

$$\mathbf{s}_{\mathcal{T}_{\text{sel}}} = \mathbf{\Phi}^T \mathbf{G} \mathbf{T}_{\text{sel}} \tag{15}$$

where $\mathbf{T}_{\text{sel}} \in \mathbb{R}_+^{|\mathcal{T}|}$ is a $|\mathcal{T}_{\text{sel}}|-$sparse vector that performs the equivalance of a masked mean computation. We then recover the standard compressed sensing problem

$$\min_{\mathbf{T}_{\text{sel}} \in \mathbb{R}_+^{|\mathcal{T}|}} \|\mathbf{T}_{\text{sel}}\|_0 \quad \text{s.t.} \quad \mathbf{\Phi}^T \mathbf{G} \mathbf{T}_{\text{sel}} = \mathbf{s}_{\mathcal{T}_{\text{sel}}} \tag{16}$$

where $\|\mathbf{T}_{\text{sel}}\|_0$ is the $\ell_0$-"norm" of $\mathbf{T}_{\text{sel}}$, *i.e.,* the number of non-zero elements in $\mathbf{T}_{\text{sel}}$. We approach this problem using the greedy orthogonal matching pursuit (OMP) algorithm and adaptively regularized iterative hard thresholding (IHT) (Axiotis & Sviridenko, 2022).

### B.4.2 Least Squares Formulation

We solve for the membership indicator via least squares. Let $\mathbf{X} \in \mathbb{R}^{N \times d}$ denote the pool features ($\mathbf{\Phi}$) and $\mathbf{Y} \in \mathbb{R}^{n \times d}$ the target gradients ($\mathbf{G}^T$). The TRAK scores satisfy

$$\mathbf{s} = \frac{1}{k} \sum_{i \in \mathcal{T}_{\text{sel}}} \mathbf{X} \mathbf{Y}_i^T = \mathbf{X} \mathbf{Y}^T \mathbf{m} \tag{17}$$

where $\mathbf{m} \in \{0, 1/k\}^n$ is the mean operator encoding membership, with $k = |\mathcal{T}_{\text{sel}}|$ non-zero entries indicating which targets are in $\mathcal{T}_{\text{sel}}$. To avoid materializing $\mathbf{X} \mathbf{Y}^T \in \mathbb{R}^{N \times n}$, we multiply both sides from the left by $\mathbf{X}^T$:

$$\mathbf{X}^T \mathbf{s} = \mathbf{X}^T \mathbf{X} \mathbf{Y}^T \mathbf{m}. \tag{18}$$

Since $\mathbf{X}^T \mathbf{X} \in \mathbb{R}^{d \times d}$ and $(\mathbf{X}^T \mathbf{X}) \mathbf{Y}^T \in \mathbb{R}^{d \times n}$ are tractable, we avoid materializing the $(N, n)$ matrix. We solve the least squares problem

$$\underset{\mathbf{m} \in \mathbb{R}^n}{\text{minimize}} \quad \|\mathbf{X}^T \mathbf{X} \mathbf{Y}^T \mathbf{m} - \mathbf{X}^T \mathbf{s}\|_2^2 \tag{19}$$

for $\mathbf{m}$. The membership scores are then given by $\mathbf{s}_{\text{lstsq}} = \mathbf{m}^*$, where $\mathbf{m}^*$ is the optimal solution.

### B.5 Combining Voting and LiRA Attacks

We analyze whether the voting-based attack for Image-based scoring and the LiRA attack provide complementary signals that can be combined for improved membership inference. The attacks exploit different aspects of the curation mechanism: voting exploits the deterministic nearest-neighbor structure, while LiRA models score distributions from shadow curation runs. Correlation analysis reveals that LiRA and voting scores have low Pearson correlation ($\rho \approx 0.3$).

We combine the two attack scores via weighted averaging:

$$s_{\text{combined}}(t) = w \cdot s_{\text{LiRA}}(t) + (1 - w) \cdot s_{\text{voting}}(t) \tag{20}$$

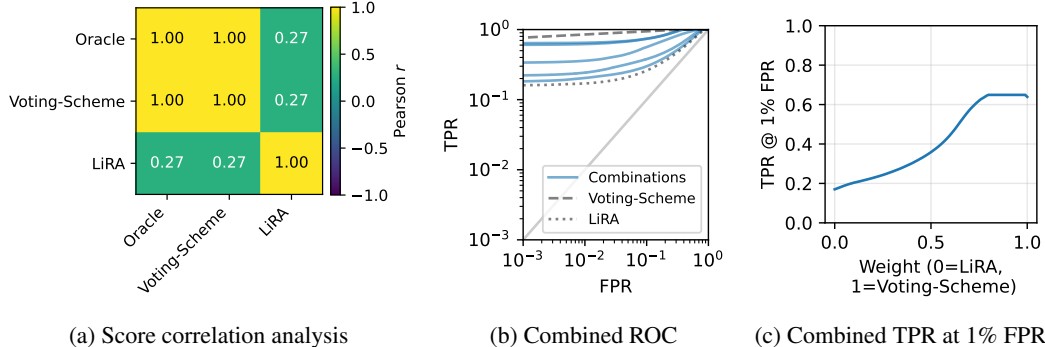

(a) Score correlation analysis      (b) Combined ROC      (c) Combined TPR at 1% FPR

Figure 9: **Combining voting and LiRA does not improve attack performance.** (a) Scatter plot showing correlation between LiRA and voting scores reveals low correlation ($\rho = 0.27$). (b) ROC curves for the combined attack and (c) TPR at 1% FPR for different weightings of the two attacks. We show that the combined attack does not improve over voting alone.

where $s_{\text{LiRA}}(t)$ and $s_{\text{voting}}(t)$ are normalized to $[0, 1]$, and $w \in [0, 1]$ controls the weighting. We evaluate $w \in \{0, 0.25, 0.5, 0.75, 1.0\}$ to identify the optimal combination.

Figure 9 shows ROC curves for the individual and combined attacks. The combined method does not improve over voting alone. Since LiRA substantially underperforms voting on Image-based scoring (AUC $\approx 0.6$ vs. $\approx 0.95$) and the voting-based attack already matches the oracle attack performance (Figure 8), LiRA contributes mostly noise when combined.

## C ATTACKING MODELS TRAINED ON CURATED DATA

### C.1 MODEL TRAINING DETAILS

All end-to-end attack experiments train CLIP models following the DataComp small-scale benchmark (Gadre et al., 2023). We provide the complete training configuration below: We use the Vision Transformer ViT-B-32 architecture (Dosovitskiy et al., 2021) for all experiments. Models are trained with the AdamW optimizer (Loshchilov & Hutter, 2019) with the following hyperparameters:

- Learning rate: $5 \times 10^{-4}$ with cosine decay schedule

- Linear warmup: 500 steps

- Weight decay: 0.2

- Beta coefficients: $\beta_1 = 0.9$, $\beta_2 = 0.98$

- Gradient clipping: maximum norm of 1.0

- Batch size: 1024

- Precision: Automatic mixed precision (AMP)

**Training Budget.** All models are trained for a fixed budget of 10M samples. The number of epochs is adjusted based on the curated pool size, *e.g.,* a pool of 100k samples results in 100 epochs, while a pool of 1M samples results in 10 epochs. This ensures all models receive equivalent amounts of training regardless of curation pool size.

**Training Data.** Models are trained exclusively on the curated subset $\tilde{\mathcal{D}}$ selected from CommonPool (small). They are *never* trained on the private target data $\mathcal{T}$ directly—they only benefit from curation performed using $\mathcal{T}$.

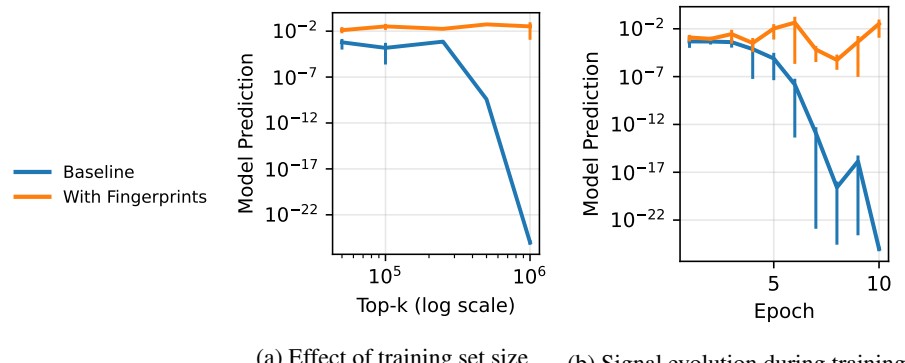

(a) Effect of training set size    (b) Signal evolution during training

Figure 10: **Mislabeled images leave a measurable fingerprint in the trained model.** We fix the number of fingerprints at $5$ and measure probability on the mislabeled concept. (a) The signal remains detectable even as the training set grows to $1{,}000{,}000$ samples ($0.0005\%$ fingerprint rate), though variance increases. Non-fingerprinted models consistently produce near-zero signal, enabling reliable fingerprint detection. (b) The fingerprint signal strengthens with additional training epochs.

## C.2  IMAGE-BASED

Image-based curation scores each pool sample $x \in \mathcal{D}$ based on its maximum similarity to any target sample:

$$s(x) = \max_{t \in \mathcal{T}} \cos(\phi(x), \phi(t)) \tag{21}$$

where $\phi(\cdot)$ is the CLIP embedding function. This creates a one-to-many correspondence mapping between targets and pool samples.

The key insight: When a target $t^*$ is included in the victim's subset $\mathcal{T}_{\text{victim}}$, all pool samples for which $t^*$ is the nearest neighbor will receive higher scores and may cross the selection threshold.

We note that by default most target samples do not affect any target samples, as Figure 2 shows. But, we assume the adversary can inject samples with embeddings $X = \{x_1, \ldots, x_M\} \subset \mathbb{R}^d$ into the pool to probe membership.

---

**Algorithm 4** Image-based Correspondence Mapping

---

**Input:** Candidate embeddings $\Phi_C$, target pool embeddings $\Phi_{\mathcal{T}}$, mixing parameter $\alpha$
**Phase 1: Find correspondences**
For each target $t \in \mathcal{T}$:
    Find $k$ candidate nearest neighbors: $\text{NN}_k(t) = \{x_1, \ldots, x_k\}$
    Compute attraction score: $a_i = \cos(\phi(t), \phi(x_i))$ (similarity to target)
    Compute repulsion score: $r_i = \max_{t' \neq t} \cos(\phi(t'), \phi(x_i))$ (similarity to other targets)
    Combined score: $s_i = (1 - \alpha) \cdot a_i + \alpha \cdot r_i$
    Select best match: $x^* = \arg\max_{x \in \text{NN}_k(t)} s_i$
    Store mapping: $\mathcal{M}(t) = x^*$
**Phase 2: Identify correspondence uniqueness**
For each selected candidate $x \in \{x^* : x^* = \mathcal{M}(t) \text{ for some } t\}$:
    Count targets mapping to it: $|\mathcal{M}^{-1}(x)| = |\{t : \mathcal{M}(t) = x\}|$
**Output:** Correspondence mapping $\mathcal{M}$ and uniqueness counts $|\mathcal{M}^{-1}(x)|$

---

For each candidate sample $x$, we compute its baseline percentile against the full target pool:

$$p_0(x) = \text{percentile}\left(\max_{t \in \mathcal{T}} \cos(\phi(x), \phi(t))\right) \tag{22}$$

This represents the expected ranking when the target randomly selects their subset $\mathcal{T}_{\text{Sel.}}$.

---

**Algorithm 5** Image-based End-to-End MIA

---

**Input:** Fingerprinted samples $\mathcal{F}$, selection observations $\{\text{selected}_x : x \in \mathcal{F}\}$, baseline percentiles $\{p_0^x\}$, selection rate $\rho$
Compute selection threshold: $\tau = (1 - \rho) \times 100$ (*e.g.*, $\pi_{50}$)
For each target $t \in \mathcal{T}$:
   Initialize: $\text{surprise}_t = 0$, $\text{count}_t = 0$
   For each fingerprinted sample $x \in \mathcal{F}$ where $t \in \mathcal{M}^{-1}(x)$:
      Compute expected selection probability:
         $P_{\exp}(x) = \frac{1}{1+\exp(-(p_0^x - \tau)/10)}$ (sigmoid model)
      Observe: $\text{selected}_x$ (whether fingerprinted sample $x$ was selected)
      Compute surprise:
         If $\text{selected}_x = \text{true}$: $\delta = 1 - P_{\exp}(x)$ (high surprise if unlikely selection)
         If $\text{selected}_x = \text{false}$: $\delta = -P_{\exp}(x)$ (negative surprise if expected selection missed)
      Weight by uniqueness: $w = 1/|\mathcal{M}^{-1}(x)|$
      Update: $\text{surprise}_t \leftarrow \text{surprise}_t + w \cdot \delta$
      Update: $\text{count}_t \leftarrow \text{count}_t + 1$
   Average: $\text{MIA}_t = \text{surprise}_t / \max(\text{count}_t, 1)$
**Output:** MIA scores $\{\text{MIA}_t\}_{t \in \mathcal{T}}$

---

The adversary then marks samples for monitoring based on information potential: For sample $x$ with targets $\mathcal{M}^{-1}(x)$:

$$\text{InfoScore}(x) = \text{CrossingScore}(x) \times \text{UniquenessScore}(x) \tag{23}$$

where:

$$\text{CrossingScore}(x) = \begin{cases} \frac{1}{1+e^{(\tau - p_0(x) - 10)/5}} & \text{if } p_0(x) < \tau \\ 0.1 & \text{if } p_0(x) \geq \tau \end{cases} \tag{24}$$

$$\text{UniquenessScore}(x) = \frac{1}{|\mathcal{M}^{-1}(x)|} \tag{25}$$

The adversary marks the top-scoring samples that are most likely to reveal membership through threshold crossing. Having stronger a-priori bounds on $\tau$, the percentile the target will use for curation, will improve this attack. In practice, we assume $\tau$ to be 50%, meaning the target will probably not pick from the worse half of public data.

Using this information, the adversary selects the highest-scoring candidates to form the fingerprinted set $\mathcal{F} \subseteq \Phi_C$. These samples are then inserted into the pool $\mathcal{D}$ with altered captions to enable detection.

Algorithm 5 shows the full algorithm.

## C.3 TRAK

The TRAK system computes influence scores $\mathbf{s} = \frac{1}{|\mathcal{T}|} \sum_{t \in \mathcal{T}} \boldsymbol{\Phi}^T \mathbf{G}_t$ where $\boldsymbol{\Phi}$ are the projected pool features and $\mathbf{G}_t$ are target gradients. However, when the target uses a secret subset $\mathcal{T}_{\text{Sel.}} \subseteq \mathcal{T}$, the actual scores become:

$$\mathbf{s}_{\mathcal{T}_{\text{Sel.}}} = \frac{1}{|\mathcal{T}_{\text{Sel.}}|} \sum_{t \in \mathcal{T}_{\text{Sel.}}} \boldsymbol{\Phi}^T \mathbf{G}_t \tag{26}$$

We assume the adversary can inject samples with gradients $C = \{c_1, \ldots, c_M\} \subset \mathbb{R}^d$ into the pool to probe membership. Obviously, gradients cannot be chosen arbitrarily. Instead, we obtain these as gradients of target samples with modified captions.

When a contrastive gradient $c_i$ is appended to the pool matrix $X$, the Sherman-Morrison formula gives its score as:

$$z_{\text{new}} = q_{\text{new}} \cdot \frac{c_i^\top G^{-1} y}{1 + c_i^\top G^{-1} c_i} \tag{27}$$

where $G^{-1} = (X^\top X + \lambda I)^{-1}$ is the regularized inverse Gram matrix and $y = \frac{1}{m} \sum_{s \in S} y_s$ is the secret average.

The key insight is that including target $y_j$ in the secret subset creates a change in scores of the samples:

$$\Delta_j(c_i) = \frac{q_{\text{new}}}{m} \cdot \frac{c_i^\top G^{-1} y_j}{1 + c_i^\top G^{-1} c_i} \tag{28}$$

This membership signal is proportional to the preconditioned inner product $c_i^\top G^{-1} y_j$ and inversely proportional to the subset size $m$, making smaller subsets more vulnerable.

Assuming target gradients have mean $\mu$ and covariance $\Sigma$, the signal-to-noise ratio for detecting target $y_j$ using contrastive $c_i$ is:

$$\text{SNR}_j(c_i) = \frac{|c_i^\top G^{-1} y_j|}{\sqrt{m \cdot c_i^\top G^{-1} \Sigma G^{-1} c_i}} \tag{29}$$

For each target $y_j$, the adversary selects the contrastive maximizing SNR:

$$i^*(j) = \arg\max_{i \in [M]} \frac{|c_i^\top G^{-1} y_j|}{\sqrt{c_i^\top G^{-1} \Sigma G^{-1} c_i}} \tag{30}$$

The attackability score $\mathcal{A}_j = \max_i \text{SNR}_j(c_i)$ quantifies how detectable each target's membership is.

he attack succeeds when the contrastive gradient crosses the curation selection threshold. Let $\tau_k$ be the score of the $k$-th ranked pool sample. The attack is successful if:

- Under $H_0$ (target not in subset): $z_{H_0} < \tau_k$ (not selected)
- Under $H_1$ (target in subset): $z_{H_1} \geq \tau_k$ (selected)

This binary change in selection status provides a clear membership signal. The attack is most effective when the contrastive score under $H_0$ is just below $\tau_k$, requiring only a small membership signal to cross the threshold.

To maximize the sensitivity to our target samples we experiment with adding copies of the target data to the pool. These should naturally have a high utility. To imprint a measurable membership signal in the model, we need to add some additional information, though. Mislabeling the samples as Carlini & Terzis (2022) did would serve that purpose, but those samples would not get picked by the TRAK algorithm, as it specifically rejects mislabeled samples as Park et al. (2023) have shown. So instead, we add *harmless and orthogonal information, i.e.,* we retain the original and useful caption and add a suffix, which mostly preserves the TRAK signal (see Section C.5). Figure 11 shows the measurable difference in models trained on such samples with additional text appended to the caption.

## C.4  COMPUTATIONAL AND QUERY COST

We analyze the computational and query costs of our proposed attacks. Unlike traditional membership inference attacks that require training on the order of hundreds of shadow models (Shokri et al., 2017; Carlini et al., 2022a), our attacks require a few curation runs, are training-free, and scale with the curation algorithms themselves.

**Computational Complexity.**  Let $N = |\mathcal{D}|$ denote the pool size, $n = |\mathcal{T}|$ the target dataset size. Our attacks require running only the respective curation algorithms or operations of similar complexity, therefore scaling linearly with the number of pool samples, as do the curation methods. For Image-based scoring, a curation costs $\mathcal{O}(Nn)$ for computing cosine similarities. For TRAK, this requires training $M$ model checkpoints and computing gradients for all $N$ pool samples, with complexity $\mathcal{O}(M \cdot N \cdot F)$ where $F$ is the number of operations in a forward pass.

- **Score-based attacks** require one curation run.
- **Subset-based attacks** require approximately 10 iterative curation runs until convergence.

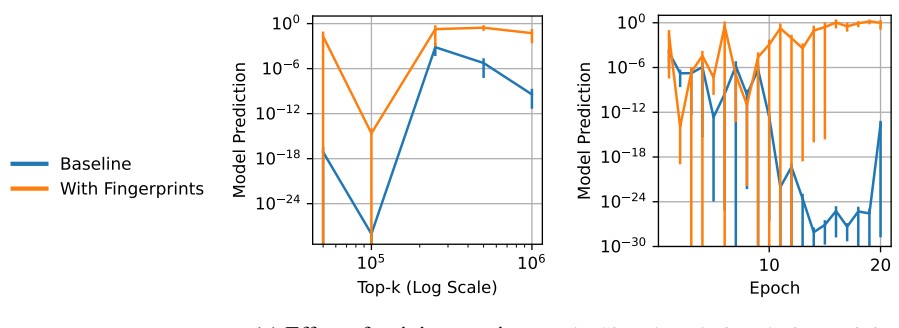

(a) Effect of training set size      (b) Signal evolution during training

Figure 11: **Samples with harmless orthogonal information imprint a measurable signal in the model.** In this case, we insert samples from CIFAR10 with an appropriate caption from CLIP-Benchmark with an additional 'and ratatouille' in the end. We then measure the model zeroshot probaability for the concept 'ratatouille'. This effect has more variance than Figure 10 shows for mislabeled images, but still distinguishes models trained with and without the fingerprints.

- **End-to-end attacks** require one curation run to identify fingerprints.

This makes our attacks practical at any scale where curation is practical in the first place.

**Query Budget.** For end-to-end black-box attacks using fingerprinting (Section C), we analyze the number of queries required to the target model.

To attack a single target sample, we need at most 5 black-box queries, one for each fingerprint sample identified during the curation phase. In practice, some fingerprints contribute to multiple targets when they share nearest-neighbor relationships. This causes the query budget to scale at most linearly with the number of targets and on expectation sublinearly with respect to $|\mathcal{T}|$.

### C.5 Caption Properties in Curation Poisoning

To understand which caption should be used for TRAK fingerprints, we analyze the effect of altering the captions on the curation score.

**Experimental Design.** We systematically ablate over three key dimensions of caption design: (1) **concept type**: correct class label baseline (ID 0), wrong in-distribution CIFAR-10 classes (IDs 1–4), and out-of-distribution concepts (IDs 5–19); and (2) **caption templates**: simple, article, imageof, photoof, and combined variants (IDs 8–13). Table 3 shows the complete set of 20 caption configurations tested. We also compare different OOD concepts (ratatouille, shader, medical scan, abstract art) and combined templates that pair an OOD concept with the correct class label from the image to understand detectability and curation signal strength trade-offs.

For each caption configuration in Table 3, we analyze gradient correlations with the baseline configuration (ID 0) to understand how caption properties impact TRAK curation behavior.

OOD concepts (IDs 5–7) with explicit templates like `photoof` and `imageof` (IDs 10–11) alter TRAK gradients too drastically, resulting in low curation signals that undermine attack effectiveness. We find that `combined` templates (IDs 12–19) offer the most practical trade-off: by pairing an OOD concept with the correct class label, their gradients correlate highly with the baseline (ID 0), preserving the TRAK curation behavior while planting detectable information in the model. This makes combined captions the optimal choice for practical curation poisoning attacks.

### C.6 Measuring Worst-Case Privacy Leakage

While the above attacks already are effective in showing privacy leakage from curation pipelines in natural setups, we further want to approximate the worst-case privacy leakage that can occur. Therefore, we rely on inserting canaries $\mathcal{C}$ into the private data $\mathcal{T}$. Canaries are sample specifically crafted

Table 3: Caption templates used in ablation study. We test variations across ID (in-distribution) and OOD (out-of-distribution) concepts using different template styles. ID 0 uses the correct class label (baseline, denoted as `class`). IDs 1–4 use wrong ID classes (*e.g.,* labeling a dog image as "airplane"). IDs 5–11 use pure OOD concepts. Combined templates (IDs 12–19) pair an OOD concept with the correct class label (*e.g.,* "dog and ratatouille" for a dog image).

| ID | Concept | Type | Template | Caption |
|----|---------|------|----------|---------|
| 0 | class | id | `photoof` | a photo of `class` |
| 1 | airplane | id | `photoof` | a photo of airplane |
| 2 | automobile | id | `photoof` | a photo of automobile |
| 3 | bird | id | `photoof` | a photo of bird |
| 4 | cat | id | `photoof` | a photo of cat |
| 5 | ratatouille | ood | `photoof` | a photo of ratatouille |
| 6 | shader | ood | `photoof` | a photo of shader |
| 7 | medical scan | ood | `photoof` | a photo of medical scan |
| 8 | ratatouille | ood | `simple` | ratatouille |
| 9 | ratatouille | ood | `article` | a ratatouille |
| 10 | ratatouille | ood | `imageof` | an image of ratatouille |
| 11 | ratatouille | ood | `photoof` | a photo of ratatouille |
| 12 | ratatouille | id + ood | `combined_imageof` | an image of `class` and ratatouille |
| 13 | ratatouille | id + ood | `combined_photoof` | a photo of `class` and ratatouille |
| 14 | shader | id + ood | `combined_imageof` | an image of `class` and shader |
| 15 | shader | id + ood | `combined_photoof` | a photo of `class` and shader |
| 16 | medical scan | id + ood | `combined_imageof` | an image of `class` and medical scan |
| 17 | medical scan | id + ood | `combined_photoof` | a photo of `class` and medical scan |
| 18 | abstract art | id + ood | `combined_imageof` | an image of `class` and abstract art |
| 19 | abstract art | id + ood | `combined_photoof` | a photo of `class` and abstract art |

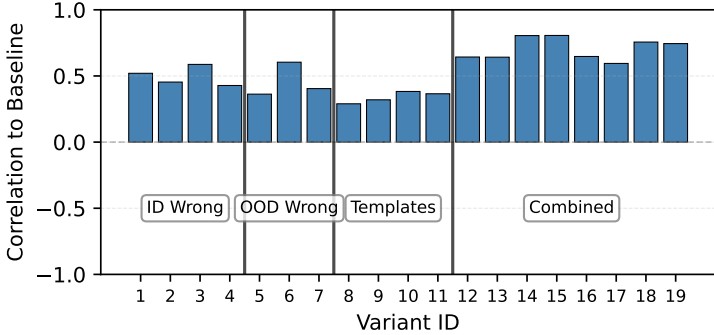

Figure 12: **Gradient correlation with baseline configuration.** Combined templates (IDs 12–19) that pair an OOD concept with the correct class label maintain the highest gradient correlation with the baseline (ID 0), preserving natural TRAK curation behavior. Other caption variants, while potentially more detectable, alter gradients too drastically and reduce attack signal strength.

to be attackable and have been used succesfully to assess empirical privacy leakage (Mahloujifar et al., 2024) and verifiy formal privacy guarantees (Annamalai et al., 2024).

For **Image-based Curation** To craft an Image-based curation canary $c$ and probe $p$, we can first select any two similar and very low-scoring (*e.g.,* $\pi_1$) samples from $\mathcal{D}$. The low score ensures that they are far away from any sample in $\mathcal{T}$, so our probe $p$ will not be selected during curation by default. With a high similarity between $p$ and $c$, we can also reliably ensure that if $c$ is inserted into $\mathcal{T}$, $p$ will score high.

For **TRAK-based Curation**, we recount that TRAK computes the score of a training sample $x \in X$ for the average $\mu$ of $T = \text{grads}(\mathcal{T})$ as $x(X^T X)^{-1}\mu q$.

$$\sigma(p) := p^\top G^{-1} \Sigma G^{-1} p.$$

Similarly to the attack in Section 3.5, we derive (in more detail in Section C.6.1) the Signal-to-Noise-Ratio (SNR) between target canary $c$ and pool sample $p$ as

$$\mathsf{SNR}(p, c) \;=\; \frac{|\Delta(p, c)|}{\sigma(p)} \;=\; \frac{|p^\top G^{-1} c|}{\sqrt{p^\top G^{-1} \Sigma G^{-1} p}}.$$

We insert those canary pairs that yield the highest SNR.

### C.6.1 CRAFTING TRAK CANARIES

TRAK computes the score of a training sample $x \in X$ for the average $\mu$ of $T = \text{grads}(\mathcal{T})$ as $x(X^T X)^{-1}\mu q$. If we append a probe $p \in \mathbb{R}^d$ to $X$, the score of that new row (using the Sherman–Morrison formula) is

$$z_{\text{new}} \;=\; \frac{p^\top G^{-1} \mu}{1 + p^\top G^{-1} p} \cdot q_{\text{new}}.$$

If we formulate as a hypothesis test whether our canary gradient $c \in \mathbb{R}^d$ is part of the target gradients $T$ ($H_1$) or is not part ($H_0$), we can define the following statistics

$$y \;=\; \mu + \varepsilon \quad (H_0), \quad \text{or} \quad y \;=\; \mu + \frac{1}{m}c + \varepsilon \quad (H_1),$$

with

$$\mathbb{E}[\varepsilon] = 0, \qquad \text{Cov}(\varepsilon) \;\approx\; \Sigma,$$

For a fixed probe $p$, we can define the baseline influence $b$, the denominator $d$ and the canary influence $c$ as

$$b(p) := p^\top G^{-1} \mu, \qquad d(p) := 1 + p^\top G^{-1} p, \qquad s(p, c) := p^\top G^{-1} c.$$

Then under $H_0$

$$s_0(p) \;=\; q_{\text{new}} \cdot \frac{b(p)}{d(p)}.$$

Under $H_1$ adding $c$ leads to a change in score for $p$ of

$$\Delta(p, c) \;=\; q_{\text{new}} \cdot \frac{s(p, c)}{m\, d(p)}.$$

The standard deviation of the probe score in either case is

$$\sigma(p) := p^\top G^{-1} \Sigma G^{-1} p.$$

### C.7 CANARY ANALYSIS

For **Image-based** curation, we manage to find canaries that will reliably leak their presence in the shaders21k dataset. For **TRAK**, our canaries achieve similar success rates as the original attack on the final trained model. We hypothesize that TRAK's resistance to canary attacks stems from the limited adversary control. Despite optimizing for maximum signal-to-noise ratio, canaries provide only marginal improvements under a realistic threat scenario. Since the canary gradients are obtain contrastively with thousands of other samples the adversary cannot control, then randomly projected into a lower-dimensional space, and afterwards averaged with all other gradients, it remains challenging to craft a strong signal.

## D MITIGATING PRIVACY LEAKAGE

We investigate how to mitigate the privacy risks outlined in our work. Section D.1 outlines how we employ DP to protect the target dataset. Section D.2 shows the effect of removing the most vulnerable samples.

## D.1 DIFFERENTIAL PRIVACY

We provide detailed methodology and experimental results for the differentially private (DP) adaptions of curation methods introduced in Section 5.

### D.1.1 DP IMAGE-BASED CURATION

In standard Image-based curation each pool sample's score is determined by its nearest neighbor in the target set. This deterministic nearest-neighbor structure creates a direct correspondence between scores and individual target samples, making the method highly vulnerable to MIAs.

**DP Noisy Max.** The most natural way to privatize the nearest-neighbor scoring is the *Report Noisy Max* mechanism (Dwork et al., 2014): for each pool sample, we compute the cosine similarity to every target embedding, add calibrated Gaussian noise to each similarity, and then take the maximum:

$$s_{\text{NM}}(x) = \max_{t \in T} \left[ \cos(\phi(x), \phi(t)) + \eta_t \right], \quad \eta_t \sim \mathcal{N}(0, \sigma^2) \tag{31}$$

Since cosine similarity is bounded in $[-1, 1]$, adding or removing a single target sample changes the set of queries by one entry with sensitivity $\Delta = 2$. The noise is calibrated as:

$$\sigma = \frac{2}{\varepsilon} \sqrt{2 \log(1.25/\delta)} \tag{32}$$

This preserves the nearest-neighbor semantics—each pool sample is still scored by its best match in the target set—while providing $(\varepsilon, \delta)$-DP.

**DP Mean.** A stronger alternative replaces the nearest-neighbor operation entirely with a distance to the DP mean of all target embeddings:

$$s_{\text{DP}}(x) = \cos(\phi(x), \bar{\phi}_T + \eta) \tag{33}$$

where $\bar{\phi}_T = \frac{1}{|T|} \sum_{t \in T} \phi(t)$ is the mean target embedding, and $\eta \sim \mathcal{N}(0, \sigma^2 I)$ is Gaussian noise calibrated via the Gaussian mechanism (Dwork et al., 2006b).

The $\ell_2$-sensitivity of the mean is:

$$\Delta_2 = \frac{1}{|T|} \cdot \max_{t, t'} \|\phi(t) - \phi(t')\|_2 \leq \frac{2}{|T|} \tag{34}$$

for normalized embeddings $\|\phi(t)\|_2 = 1$. To achieve $(\varepsilon, \delta)$-DP, we set:

$$\sigma = \frac{\Delta_2}{\varepsilon} \sqrt{2 \log(1.25/\delta)} = \frac{2}{|T|\varepsilon} \sqrt{2 \log(1.25/\delta)} \tag{35}$$

**Comparison.** Table 5 compares both approaches. The DP mean provides stronger privacy at equivalent $\varepsilon$ because averaging already obscures individual target contributions, while the noisy max retains the per-target nearest-neighbor structure. However, the noisy max may be preferable when the nearest-neighbor semantics are important for downstream curation quality.

### D.1.2 DP TRAK CURATION

TRAK-based curation computes scores via $s(x) = \Phi(x)^T \bar{g}$, where $\bar{g} = \frac{1}{|T|} \sum_{t \in T} G_t$ is the mean gradient over target samples. While averaging provides some natural privacy protection, small target sets ($|T| < 5000$) remain highly vulnerable to membership inference attacks.

We privatize the mean gradient computation:

$$\bar{g}_{\text{DP}} = \frac{1}{|T|} \sum_{t \in T} \tilde{G}_t + \eta, \quad \eta \sim \mathcal{N}(0, \sigma^2 I) \tag{36}$$

where $\tilde{G}_t$ are the clipped gradients

$$\tilde{G}_t = G_t \cdot \min \left( 1, \frac{C}{\|G_t\|_2} \right) \tag{37}$$

with clipping threshold $C$ and sensitivity $\Delta_2 = 2C/|T|$. The noise scale is $\sigma = \frac{2C}{|T|\varepsilon} \sqrt{2 \log(1.25/\delta)}$.

Table 4: **Attack success on curation scores for DP Image-based and TRAK curation.**

| Privacy Guarantee | TPR @ 1% FPR | |
|---|---|---|
| $(\varepsilon, \delta = 1e-5)$ | Image-Based Curation | TRAK |
| non-private | $0.9842 \pm 0.0013$ | $1.0000 \pm 0.0000$ |
| 1000 | $0.0160 \pm 0.0000$ | $1.0000 \pm 0.0000$ |
| 100 | $0.0155 \pm 0.0000$ | $0.3324 \pm 0.0797$ |
| 10 | $0.0134 \pm 0.0000$ | $0.0165 \pm 0.0072$ |
| 5 | $0.0107 \pm 0.0000$ | $0.0136 \pm 0.0071$ |
| 2 | $0.0105 \pm 0.0000$ | $0.0125 \pm 0.0065$ |

Table 5: **Impact of Nearest-Neighbor vs. Mean on the Image-Based Privacy Leakage.**

| Privacy Guarantee | TPR @ 1% FPR | |
|---|---|---|
| $(\varepsilon, \delta = 10^{-5})$ | Image-Based Curation (Mean) | Image-Based Curation (Nearest Neighbor) |
| non-private (inf) | $0.0160 \pm 0.0067$ | $0.9842 \pm 0.0013$ |
| 1000 | $0.0160 \pm 0.0069$ | $0.8591 \pm 0.0158$ |
| 100 | $0.0155 \pm 0.0113$ | $0.0542 \pm 0.0090$ |
| 10 | $0.0134 \pm 0.0069$ | $0.0110 \pm 0.0050$ |
| 5 | $0.0107 \pm 0.0065$ | $0.0109 \pm 0.0057$ |
| 2 | $0.0105 \pm 0.0045$ | $0.0106 \pm 0.0057$ |

### D.1.3 EVALUATION

We test $\varepsilon \in \{0.1, 0.5, 1.0, 2.0, 5.0, 10.0\}$ with $\delta = 10^{-5}$. We attack the curation scores with LiRA, fitted on shadow curation scores with no DP.

Table 4 shows the attack success on the curation scores for DP Image-based and TRAK curation. The results show that these measures are effective at mitigating privacy leakage. For Image-Based Curation, just replacing the nearest neighbor operation with a mean drastically reduces the attack success. For TRAK-based curation we saw higher attack success, despite using a mean, because the gradient projection dimension (32,768) is significantly higher than the embedding dimension for Image-based curation (768). This aligns with our ablation results for TRAK, where the attack success drops for less than 2,048 dimensions. This explains why even the non-private mean is hard to attack, and subsequently why a DP guarantee of $\varepsilon = 1000$ shows low attack success. Table 5 analyzes this further, comparing the mean and nearest neighbor-based curation for various privacy guarantees. The results show that the mean is harder to attack than the nearest neighbor-based curation for various privacy guarantees.

## D.2 REMOVING THE MOST VULNERABLE SAMPLES

We detail our approach and results of removing the most vulnerable samples from the target dataset.

We conduct our experiments on the curation scores for CIFAR-10. For the **TRAK-based** analysis, we use a target dataset size of 25,000 samples and employ the least squares attack. The attack is evaluated over 16 seeds. For the **Image-based** analysis, we use a target dataset size of 5,000 samples and employ the LiRA attack. The attack is evaluated over 16 seeds with 256 shadow models. In both experiments, we define the "vulnerable" set as the top 5% of samples with the highest membership inference attack success rate. For comparison, we also remove 5% of samples at random. We report AUC only on the remaining samples.

We then evaluate four scenarios:

- **Baseline**: Full target dataset.

- **Ideal**: Post-hoc removal of vulnerable samples from ROC computation.

- **Vulnerable Removal**: Re-run experiment after removing vulnerable samples from target dataset.

- **Random Removal**: Re-run experiment after removing random samples from target dataset.

**Findings.** We investigate whether removing the most vulnerable samples from the target dataset could serve as an effective defense against membership inference. The results show that for both methods, removing samples increases overall attack success.

For **Image-based** curation, Figure 13a shows that the attack success increases significantly when removing the most vulnerable samples. Removing samples at random increases the attack success only marginally. The 5% most vulnerable target samples are the score-determining nearest neighbors for 31.1% of the pool. Their removal exposes 1.8% of previously shielded targets and increases vulnerability for over 80% of target samples, which are now on average the nearest neighbor for 83 additional pool samples. Therefore, removing the most vulnerable samples actually exposes many previously protected samples. This is a strong case of the *Privacy Onion effect* (Carlini et al., 2022b).

For **TRAK-based** curation, Figure 13b shows that removing vulnerable samples increases overall attack success similarly to removing random samples. Removing the most vulnerable samples (AUC 0.9400) reduces attack success marginally more than removing random samples (AUC 0.9439). But our ablations have shown that the attack success is very sensitive to target dataset size (Fig. 34-35 in Appendix E.5), so the reduction in target dataset size outweighs the benefit of removing any samples.

Our findings suggest that simple sample removal is not a robust defense strategy. For image-based curation, the onion effect negates the benefit of removing outliers. For TRAK-based curation, while no onion effect is present, the attack remains effective, indicating that privacy leakage is not confined to a small subset of "vulnerable" points but is a systemic property of the curation process.

# E EXTENDED EVALUATION

## E.1 IMAGE-BASED SCORING ATTACKS

We show the results of the attack on Image-based scoring in Figure 14. LiRA exhibits strong and consistent performance across all datasets. The custom attack is better for all datasets at FPR $> 10^{-4}$. For lower FPRs, performance is highly variable. Figure 15 shows the results of the attack on Image-based scoring without access to the scores. The custom attack and LiRA achieve similar results, with the custom attack being slightly more successful in the low FPR region.

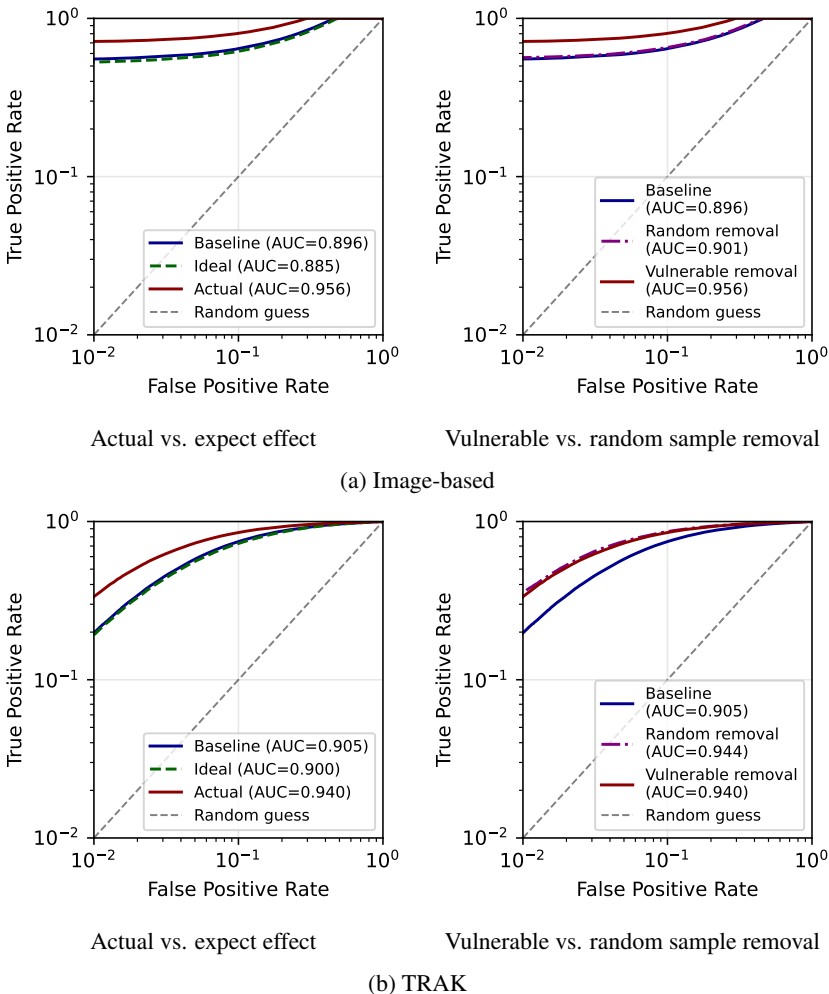

Figure 13: **Removing the most sensitive samples does not prevent leakage.** We show attack success *increases* when removing the most vulnerable samples. a) **Image-based** curation shows a strong *privacy onion effect*. Removing the most vulnerable samples increases attack success significantly, while removing random samples has a negligible effect. b) **TRAK-based** curation shows no privacy onion effect, but since attack success is highly sensitive to target dataset size, removing samples increases it.

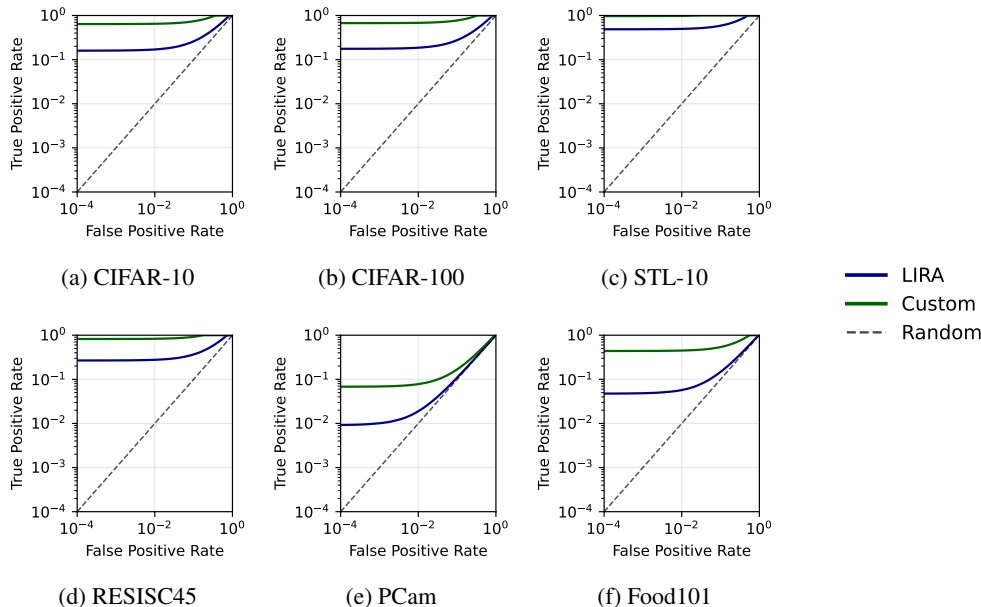

Figure 14: **Image-based Attack success with access to the scores** for various datasets and curation methods given access to the full set of pool scores. We find that - while none of the methods satify DP - only Image-based scoring is attackable.

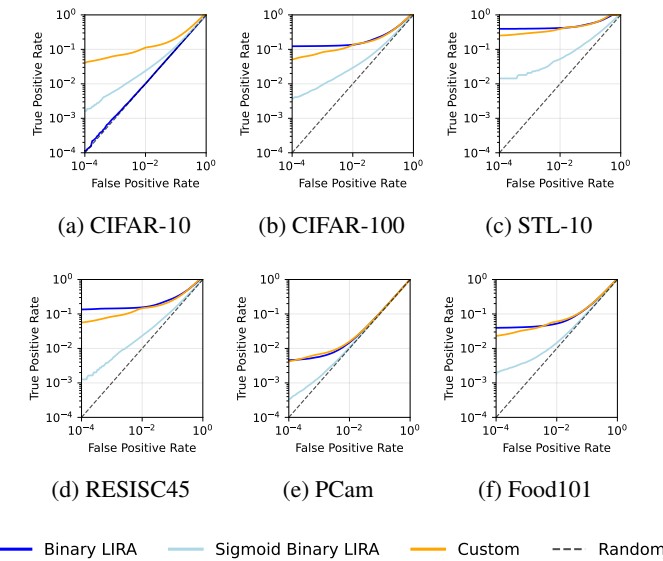

Figure 15: **Image-based Attack success without access to the scores** for various datasets and curation methods given access to the full set of pool scores. We show that just the selection of the pool samples reveals membership information about the targets.

## E.2 TRAK SCORING ATTACKS

Figures 16 and 17 show that TRAK is hardly attackable, both with access to the scores and without.

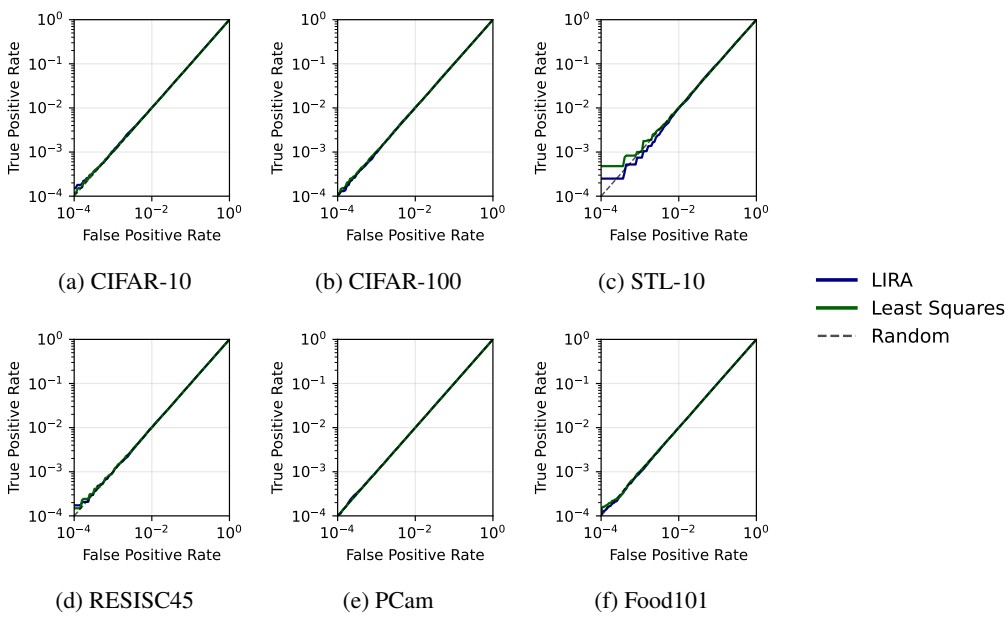

Figure 16: **TRAK Attack success with access to the scores.**

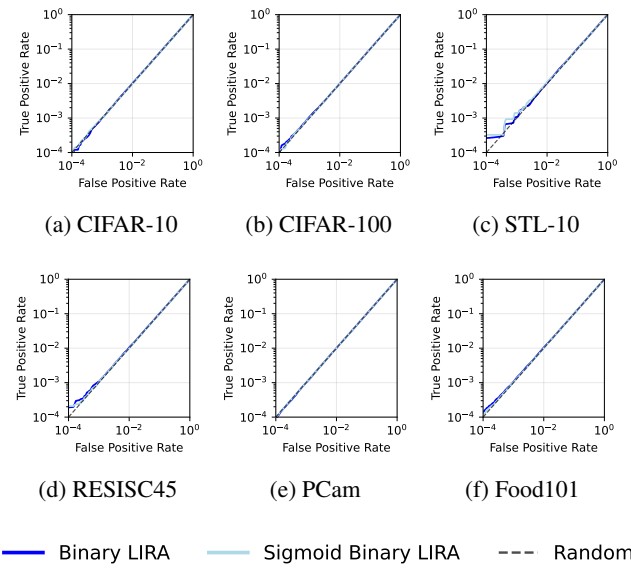

Figure 17: **TRAK Attack success without access to the scores.**

### E.3 END-TO-END ATTACKS

Figure 18 shows that models trained on Image-based curation leak membership information over all target dataset sizes. Figure 19 shows that for small target datasets, even models curated with TRAK are highly attackable, more so than Image-based curation. This success diminishes as the target dataset size is increased though.

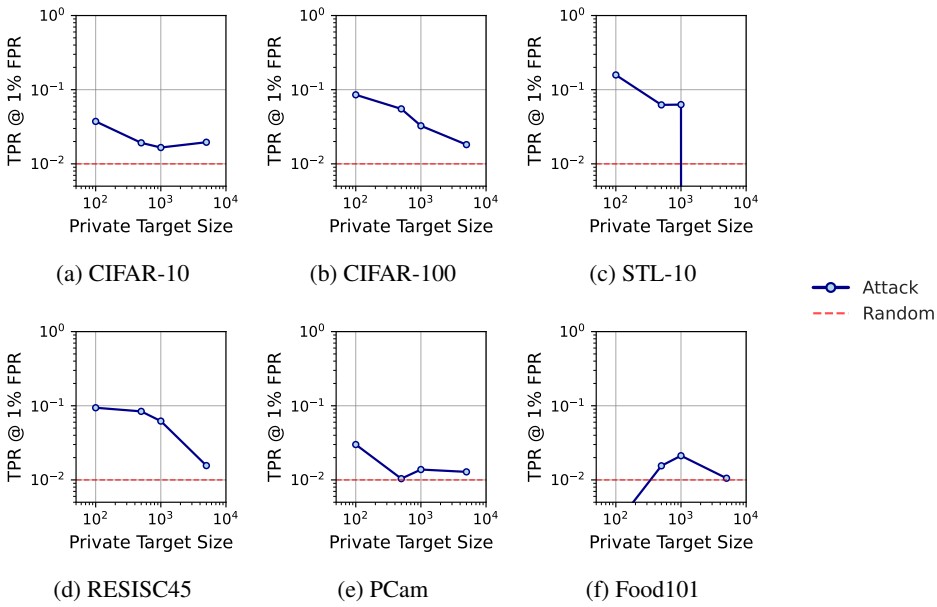

Figure 18: **Image-based attack success with access only to the models trained on curated data.**

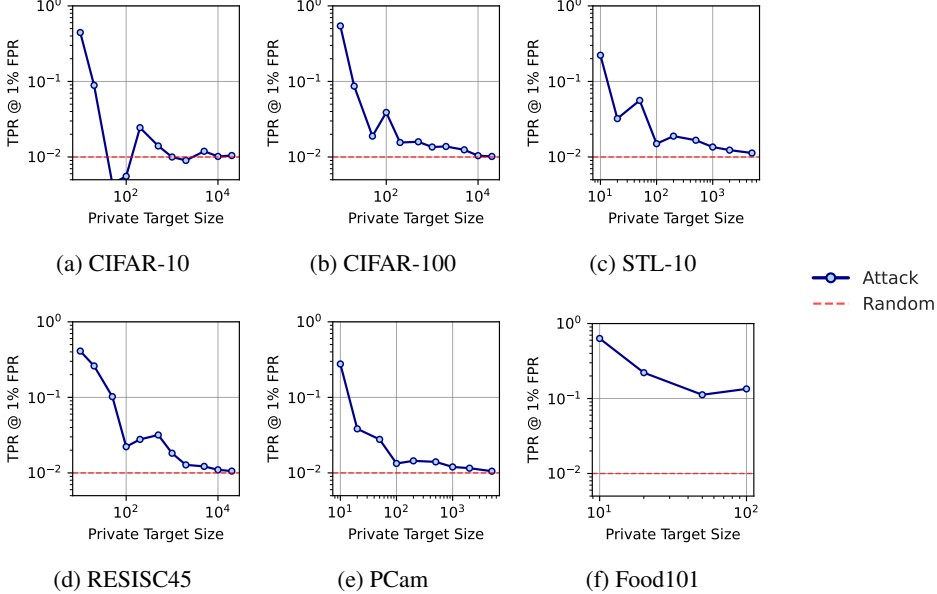

Figure 19: **TRAK attack success with access only to the models trained on curated data.**

### E.4 DETAILED ROC CURVES

Figures 20 to 31 show full ROC curves for 36 configurations of datasets and target dataset sizes for the end-to-end attack, showing the full TPR-FPR tradeoff. We furthermore add AUC plots, to enhance comparison with other work that has reported these numbers. These results show that, when the attack is successful, it succeeds across a wide range of the trade-off curve.

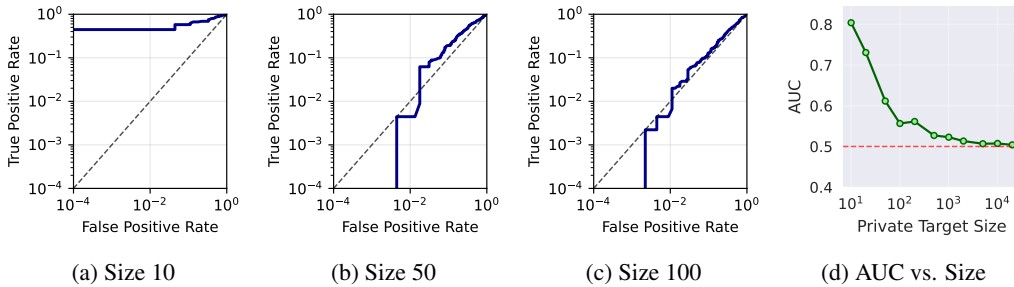

(a) Size 10     (b) Size 50     (c) Size 100     (d) AUC vs. Size

Figure 20: **TRAK end-to-end attack on CIFAR-10:** ROC curves for different target dataset sizes and AUC vs. size summary.

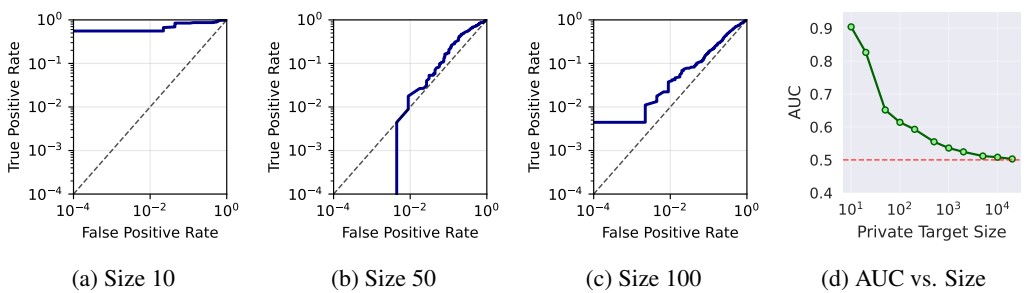

(a) Size 10     (b) Size 50     (c) Size 100     (d) AUC vs. Size

Figure 21: **TRAK end-to-end attack on CIFAR-100:** ROC curves for different target dataset sizes and AUC vs. size summary.

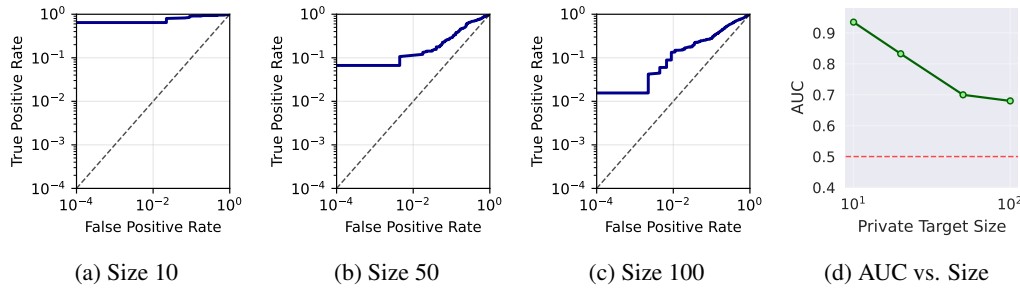

(a) Size 10     (b) Size 50     (c) Size 100     (d) AUC vs. Size

Figure 22: **TRAK end-to-end attack on Food101:** ROC curves for different target dataset sizes and AUC vs. size summary.

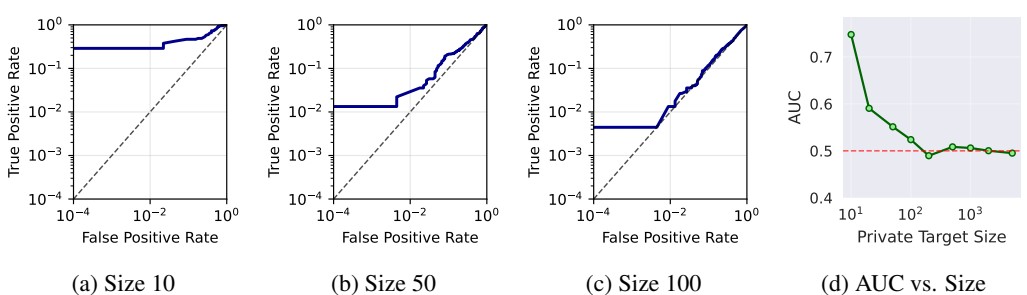

(a) Size 10     (b) Size 50     (c) Size 100     (d) AUC vs. Size

Figure 23: **TRAK end-to-end attack on PCam:** ROC curves for different target dataset sizes and AUC vs. size summary.

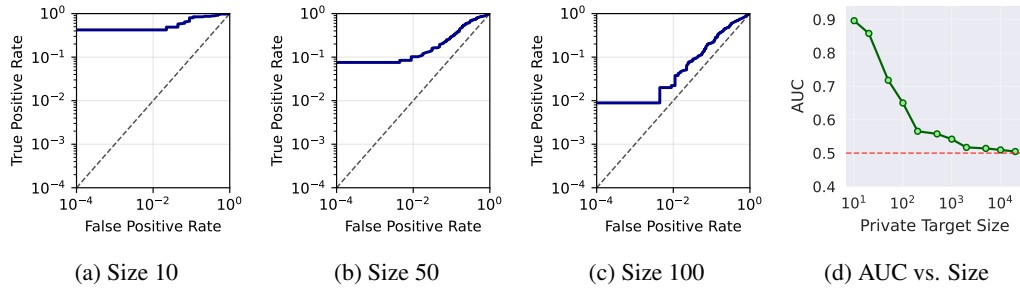

Figure 24: **TRAK end-to-end attack on RESISC45:** ROC curves for different target dataset sizes and AUC vs. size summary.

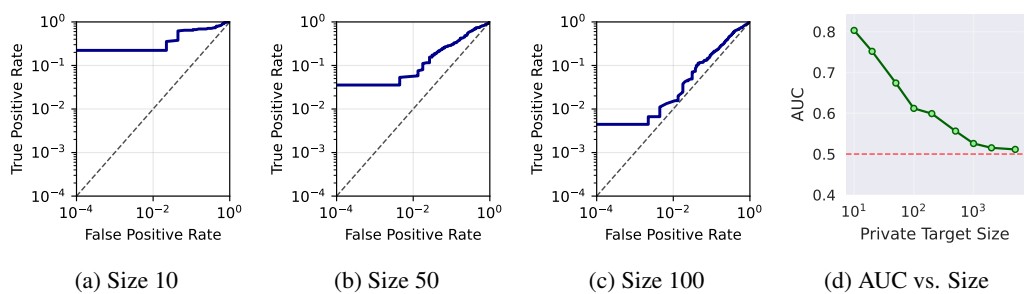

Figure 25: **TRAK end-to-end attack on STL-10:** ROC curves for different target dataset sizes and AUC vs. size summary.

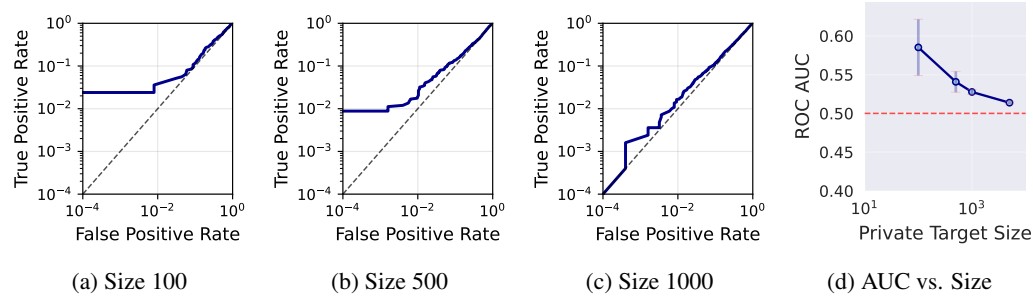

Figure 26: **Image-based end-to-end attack on CIFAR-10:** ROC curves for different target dataset sizes and AUC vs. size summary.

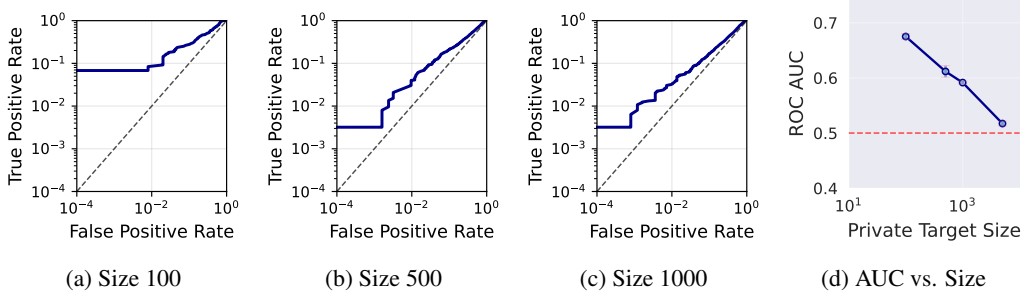

Figure 27: **Image-based end-to-end attack on CIFAR-100:** ROC curves for different target dataset sizes and AUC vs. size summary.

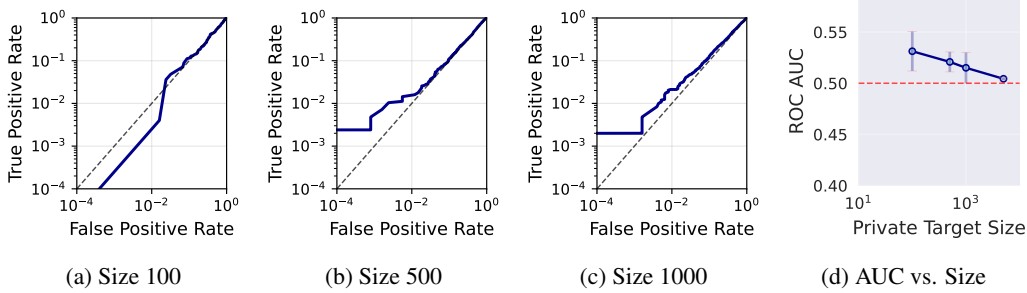

(a) Size 100    (b) Size 500    (c) Size 1000    (d) AUC vs. Size

Figure 28: **Image-based end-to-end attack on Food101:** ROC curves for different target dataset sizes and AUC vs. size summary.

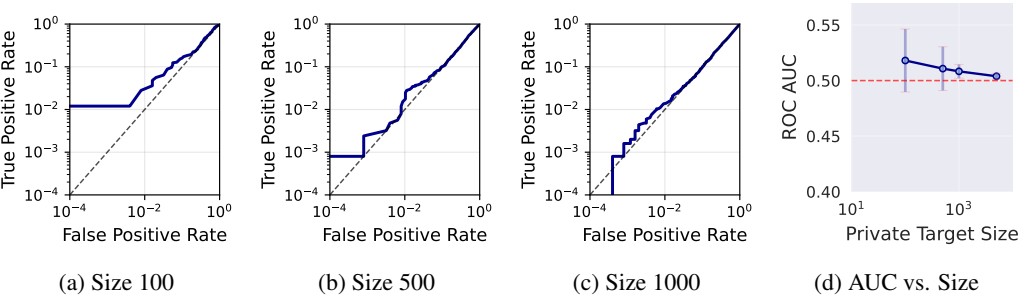

(a) Size 100    (b) Size 500    (c) Size 1000    (d) AUC vs. Size

Figure 29: **Image-based end-to-end attack on PCam:** ROC curves for different target dataset sizes and AUC vs. size summary.

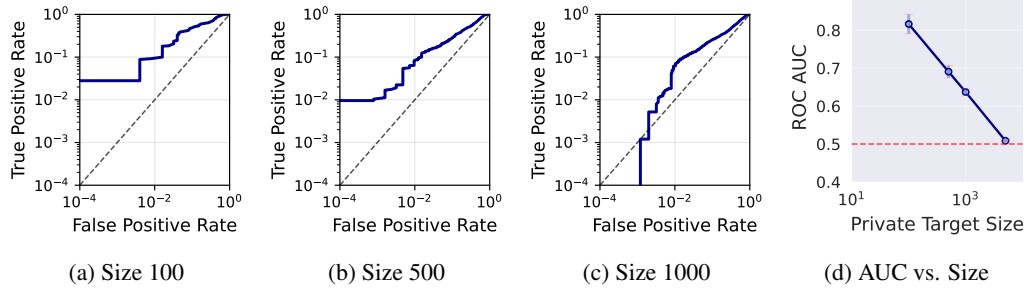

(a) Size 100    (b) Size 500    (c) Size 1000    (d) AUC vs. Size

Figure 30: **Image-based end-to-end attack on RESISC45:** ROC curves for different target dataset sizes and AUC vs. size summary.

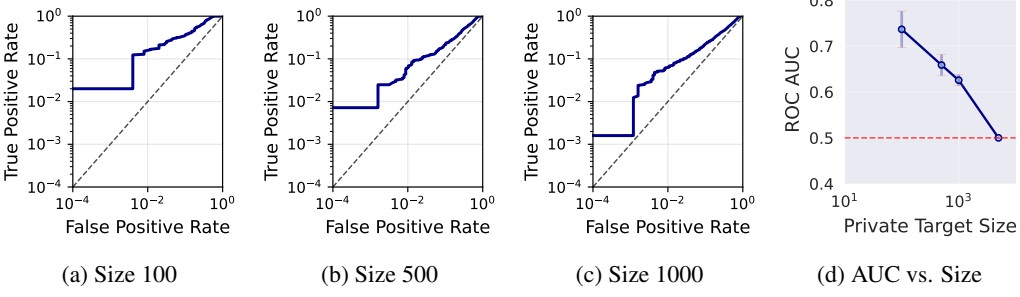

(a) Size 100    (b) Size 500    (c) Size 1000    (d) AUC vs. Size

Figure 31: **Image-based end-to-end attack on STL-10:** ROC curves for different target dataset sizes and AUC vs. size summary.

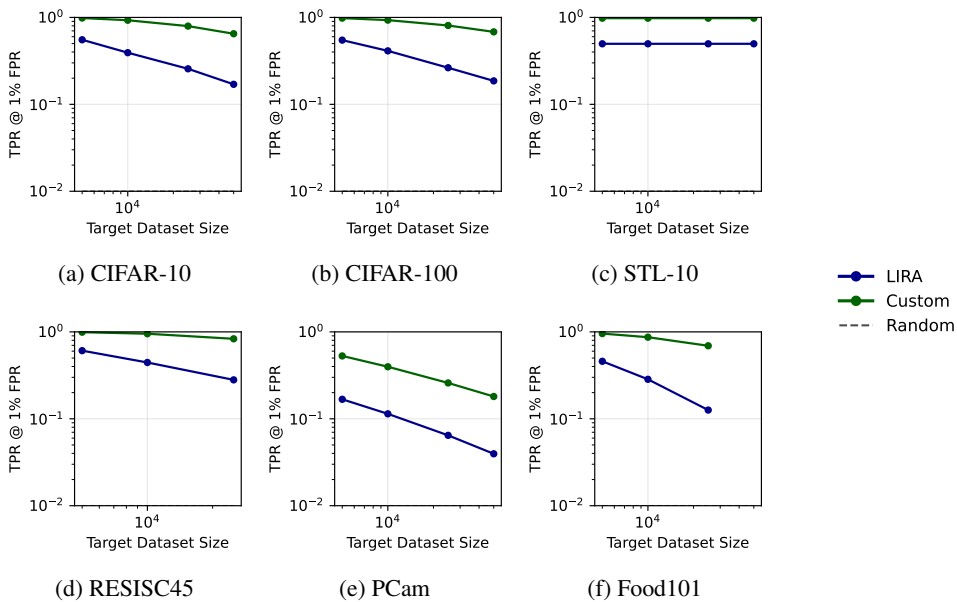

Figure 32: **Image-based attack success with access to scores** ablated over target dataset size.

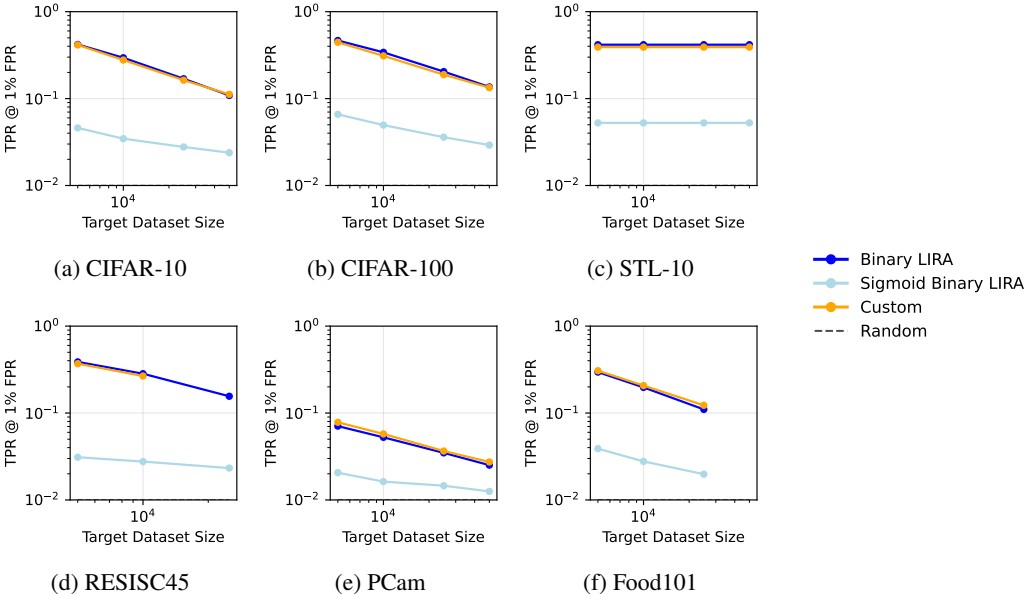

Figure 33: **Image-based attack success without access to scores** ablated over target dataset size.

### E.5    TARGET DATASET SIZE ABLATION

Figures 32 to 35 show the effect of target dataset size on attack success. For Image-based curation, the leakage is considerable at all dataset sizes. Since only the nearest neighbors determine the scores, a larger dataset size means fewer samples are exposed, but the sensitivity of the exposed samples remains constant. For TRAK-based curation, at small dataset sizes all samples are exposed, but at larger sizes the averaging has a shielding effect for all samples as well.

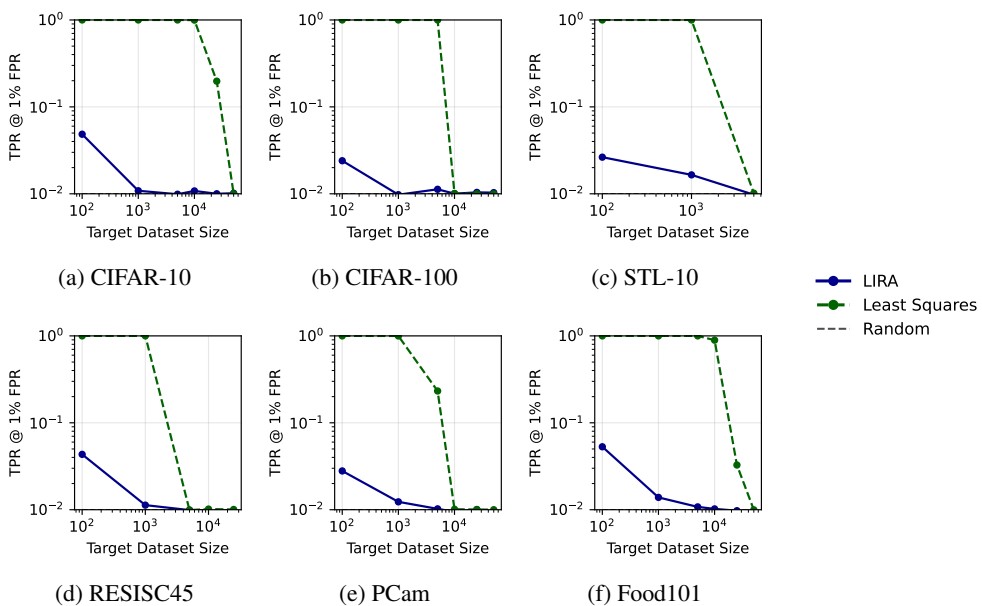

Figure 34: **TRAK-based attack success with access to scores** ablated over target dataset size.

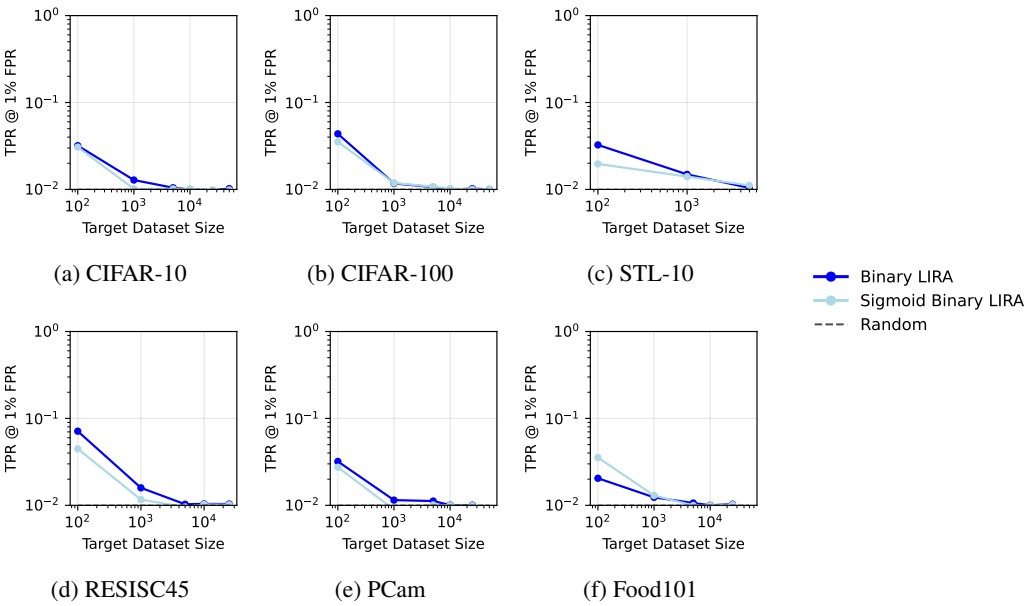

Figure 35: **TRAK-based attack success without access to scores** ablated over target dataset size.

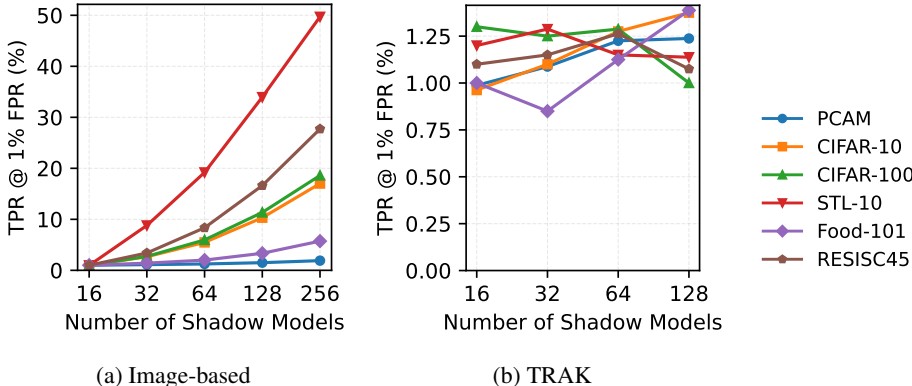

(a) Image-based         (b) TRAK

Figure 36: **Shadow models ablation.** (a) For Image-based curation we show that the improvement of adding more shadow models varies drastically between datasets. (b) For TRAK, we do not observe any improvement in attack success when adding more shadow models.

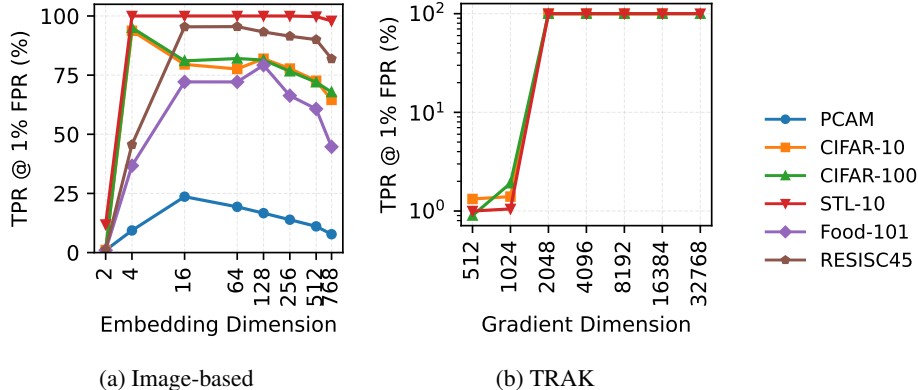

(a) Image-based         (b) TRAK

Figure 37: **Dimension ablation.** We show that the number of (a) embedding dimensions for Image-based curation and (b) projection dimensions for TRAK have a moderate effect on attack success on the scores.

## E.6 SHADOW MODELS ABLATION

Figure 36 shows that the improvement of adding more shadow models varies drastically between datasets. For Image-based curation, the improvement is connected to the sparsity of the signals that Figure 2 shows. For TRAK, the improvement is more consistent across datasets.

## E.7 DIMENSION ABLATION

Figure 37 shows the effect of changing the number of dimensions. For Image-Based curation, we show that the attack success is highest for 128 dimensions using our voting based attack (for LiRA 64 dimensions). Because attacks on Image-based curation exploit one-to-one mappings of measurements between target and pool samples, having fewer dimensions can increase the reliability of those measurements. Only for extremely few dimensions (*e.g.,* two) does the attack success drop to that of random guessing. For TRAK, we need a sufficient number of dimensions ($\leq 1024$), below which the attack success drops significantly.

## F LLM USAGE

We have used LLMs to speed up the process of typesetting figures, algorithms, and formulas in LaTeX. We further employed LLM-based agents for writing code, especially to generate figures.

