# OpenReview forum: "Curation Leaks: Membership Inference Attacks against Data Curation for Machine Learning"
_ICLR.cc/2026/Conference — ICLR 2026 Poster_

### Official Review · Reviewer_hymw · 2025-10-28

**Soundness:** 3
**Presentation:** 3
**Contribution:** 2
**Rating:** 6
**Confidence:** 4

**Summary:**

This paper uncovers a novel and important privacy vulnerability in machine learning pipelines: membership leakage during data curation. The authors demonstrate both theoretically and empirically that data curation, even before model training, can expose sensitive information about which samples were part of the raw dataset.

**Strengths:**

1. The paper addresses a new angle on privacy leakage. Most membership inference works focus on model outputs; This work focus shifts to data preprocessing and curation.
2. The problem is well motivated and clearly defined.
3. The empirical results are convincing and well-presented.

**Weaknesses:**

1. While the proposed curation leak is novel and conceptually important, its real-world applicability is limited by the assumption that the adversary can access or infer the curation outcome. In many production pipelines, the raw uncurated data or discarded samples are not observable.
2. The paper briefly mentions DP sampling and randomization but does not propose new defense mechanisms.
3. Ablation studies (e.g., effect of dataset size or embedding dimension) are not included.

**Questions:**

1.Clarify the assumption
2.Add necessary ablation studies.

---

> ### Author Response · Authors · 2025-11-21
> **New Experiments with Differential Privacy Mitigations, Ablations and Expanded Motivation**
>
> **\> While the proposed curation leak is novel and conceptually important, its real-world applicability is limited by the assumption that the adversary can access or infer the curation outcome. In many production pipelines, the raw uncurated data or discarded samples are not observable.**
>
> We agree that the attacks only on the trained models are perhaps most relevant. Still, we have extended the motivation in the introduction where we outline the following aspects: Curated datasets are often released to the public \[1-3\]. In some cases, even the quality scores *are* released \[4\]. Furthermore, our work on the leakage of the intermediate data is relevant for real life challenges in dictating organization data access, i.e., who should be allowed to access which parts of the pipeline, which is not trivial to determine \[5\]. There also exists a growing market of data curation as a service, where datasets, scores and subsets are passed around (e.g., Scale AI, Snorkel AI, DatologyAI, Cleanlab). For all of these real-life cases, it is important to have some idea of the privacy implications of what is being passed around. Our work is the first to show the risks for two state-of-the-art methods on all relevant threat levels (scores, subsets, models).
>
> \> **The paper briefly mentions DP sampling and randomization but does not propose new defense mechanisms.**
>
> Following the reviewer’s comment, we updated our paper to include adaptations of DP curation mechanisms and added experiments to show how this impacts the privacy leakage. For Image-Based Curation, we compute the scores based on the DP-Mean of the target embeddings. For TRAK, we compute the scores using a DP-Mean of the gradients. We include a table with the results on CIFAR10 below.
>
> **Ablation on the Privacy Guarantee**
>
> |Privacy ($\\varepsilon,\\delta=10^{-5}$)|TPR at 1% FPR||
> |:--|:--|:--|
> ||Image-BasedCuration|TRAK|
> |non-private|0.9842±0.0013|1.0000±0.0000|
> |1000|0.0160±0.0069|1.0000±0.0000|
> |100|0.0155±0.0113|0.3324±0.0797|
> |10|0.0134±0.0069|0.0165±0.0072|
> |5|0.0107±0.0065|0.0136±0.0071|
> |2|0.0105±0.0045|0.0125±0.0065|
>
> The results show that these measures are effective at mitigating privacy leakage. For Image-Based Curation, just replacing the nearest neighbor operation with a mean drastically reduces the attack success. Therefore, even $\\varepsilon=1000$ has a much lower TPR than the non-private variant. We added this insight to a new section 5 on **Mitigating Privacy Leakage** with details of the algorithms added to a new Appendix D.
>
> \> **Ablation studies (e.g., effect of dataset size or embedding dimension) are not included.**
>
> Based on the reviewer’s suggestion, we performed additional ablation experiments for all datasets, curation methods and attacks over the dataset size. For the Image-Based curation, we also ablate over embedding dimensions and for TRAK, we ablate over the projection dimensions. We added the full ablation as Figures 20-35 to Appendix E. To keep this response short, we present here just the results on CIFAR10 when attacking the scores.
>
> **Ablation on the Embedding / Projection Dimension**
>
> ||Embedding/Projection Dimension|TPR at 1% FPR|
> |:----|:----|:----|
> |**Image-Based**|768|0.6455±0.0036|
> ||512|0.7261±0.0029|
> ||256|0.7777±0.0027|
> ||128|0.8194±0.0022|
> ||64|0.7760±0.0021|
> ||16|0.7954±0.0083|
> ||4|0.9374±0.0017|
> ||2|0.0110±0.0007|
> |**TRAK**|32,768|1.0000±0.0000|
> ||16,384|1.0000±0.0000|
> ||8,192|1.0000±0.0000|
> ||4,096|1.0000±0.0000|
> ||2,048|1.0000±0.0000|
> ||1,024|0.0140±0.0000|
> ||512|0.0132±0.0000|
>
> **For Image-Based curation**, we show that the attack success is highest for 128 dimensions using our voting based attack (for LiRA 64 dimensions). Because attacks on Image-based curation exploit one-to-one mappings of measurements between target and pool samples, having fewer dimensions can increase the reliability of those measurements. Only for extremely few dimensions (e.g., two) does the attack success drop to that of random guessing. **For TRAK-based curation**, we need a sufficient number of dimensions ($\\leq 1024$), below which the attack success drops significantly.
>
> **Ablation on Target Dataset Size**
>
> ||Target Dataset Size|TPR at 1% FPR|
> |:----|:----|:----|
> |**Image-Based**|5,000|0.9842±0.0013|
> ||10,000|0.9284±0.0030|
> ||25,000|0.7946±0.0029|
> ||50,000|0.6491±0.0035|
> |**TRAK**|5,000|1.0000±0.000|
> ||10,000|0.9964±0.0090|
> ||25,000|0.1976±0.0234|
> ||50,000|0.0100±0.0004|
>
> **For Image-based curation**, the leakage is considerable at all dataset sizes. Since only the nearest neighbors determine the scores, a larger dataset size means fewer samples are exposed, but the sensitivity of the exposed samples remains constant. **For TRAK-based curation**, at small dataset sizes all samples are exposed, but at larger sizes the averaging has a shielding effect for all samples as well.
>
> If the above responses address the Reviewer's concerns, we would greatly appreciate it if they could adjust their score accordingly.

---

> > ### Author Response · Authors · 2025-11-21
> > **References for Our Previous Reply**
> >
> > **References**:
> >
> > [1]: Li, Z. et al. LAION-SG: An enhanced large-scale dataset for training complex image-text models with structural annotations. (2024).
> >
> > [2]: Penedo, Guilherme, et al. "The fineweb datasets: Decanting the web for the finest text data at scale." Advances in Neural Information Processing Systems 37 (2024): 30811-30849.
> >
> > [3]: Penedo, Guilherme, et al. "The refinedweb dataset for falcon llm: Outperforming curated corpora with web data only." Advances in Neural Information Processing Systems 36 (2023): 79155-79172
> >
> > [4]: Together Computer, RedPajama: an open dataset for training large language models. (2023).
> >
> > [5]: Sushko, Y. et al. Our Approach to Protecting AI Training Data. (2025).

---

> > ### Comment · Reviewer_hymw · 2025-11-24
> >
> > Thanks for the response.
> > Just one more question: for the Ablation on the Privacy Guarantee table,
> > $\epsilon$ = 1000 is a very small DP noise, but it makes a huge difference compared with the non-private settings. Could you further explain why this is happening?

---

> > > ### Author Response · Authors · 2025-11-27
> > > **Because We Replace Nearest-Neighbor with Mean**
> > >
> > > To mitigate the privacy leakage, we replace the distance to the nearest neighbor with the distance to the **mean of the target dataset**. Therefore, in the column "Image-based Curation", the row "non-private" refers to the nearest neighbor-based algorithm, where as "$\varepsilon=1000$" refers to the mean-based algorithm.
> > >
> > >  We present in the table below separate columns for both algorithms. In the left column, we show that just using the mean reduces the LiRA attack success drastically, even **without any DP guarantee**. In the right column, we added results of **further experiments**, where we keep the nearest neighbor mechanism, but employ a noisy max release mechanism.
> > >
> > > **Impact of Nearest-Neighbor vs. Mean on the Privacy Leakage**
> > >
> > > | Privacy Guarantee ($\\varepsilon,\\delta=10^{-5}$) | TPR @ 1% FPR |  |
> > > | :---- | :---- | :---- |
> > > |  | Image-Based Curation (Mean) | Image-Based Curation (Nearest Neighbor) |
> > > | non-private ($\\inf$) | 0.0160 ± 0.0067 | 0.9842 ± 0.0013 |
> > > | 1000 | 0.0160 ± 0.0069 | 0.8591 ± 0.0158 |
> > > | 100 | 0.0155 ± 0.0113 | 0.0542 ± 0.0090 |
> > > | 10 | 0.0134 ± 0.0069 | 0.0110 ± 0.0050 |
> > > | 5 | 0.0107 ± 0.0065 | 0.0109 ± 0.0057 |
> > > | 2 | 0.0105 ± 0.0045 | 0.0106 ± 0.0057 |
> > >
> > > For TRAK-based curation we saw higher attack success, despite using a mean, because the gradient projection dimension (32,768) is significantly higher than the embedding dimension for Image-based curation (768). The low attack success therefore aligns with our ablation results for TRAK, where the attack success drops for less than 2,048 dimensions. This explains why even the non-private mean is hard to attack, and subsequently why a DP guarantee of $\varepsilon=1000$ shows low attack success.

---

### Official Review · Reviewer_zhSF · 2025-10-31

**Soundness:** 2
**Presentation:** 2
**Contribution:** 2
**Rating:** 2
**Confidence:** 3

**Summary:**

The paper argues that “privacy via curation” (only using sensitive data to select public data and then training solely on the public subset) is not automatically private. They design membership inference attacks against: (i) curation scores; (ii) the selected public subset; and (iii) the final model (via a small number of crafted public “fingerprint” samples).The results show that image-embedding/nearest-neighbor style curation is much more vulnerable; TRAK-style gradient-averaged curation is more robust but still leaky for small target sets.

**Strengths:**

The problem itself seems interesting.

**Weaknesses:**

1. I’m familiar with LiRA and recognize you use an online variant, but the three attack surfaces are hard to follow because the threat model isn’t explicitly stated up front. Please spell out—on one page—(i) the adversary’s goal (membership in the private target set used for curation vs. classic training-set membership), (ii) what the adversary can observe at each stage (scores, selection mask, final model), and (iii) what the adversary can do (e.g., can they inject public items?). A single “who-sees-what” diagram for the 3 stages would save readers a lot of guesswork. Also, there are many typos in the main text and appendix; a thorough copy edit would help.

2. The paper does not convincingly explain why leakage at the scoring and subset-selection stages matters when, in many realistic deployments, only the final model is exposed. In such cases, standard membership inference on the trained model already demonstrates risk. The authors should clarify concrete scenarios (and access assumptions) where Sections 3.1 and 3.2 introduce new or additional threat surface beyond what model-only exposure already entails.

3. The end-to-end attack hinges on the ability to insert crafted samples into the public pool. This reads as logically circular: relying on an existing injection threat to establish a new leakage threat. The paper should justify the realism and scope of this assumption (e.g., where such insertion is feasible, at what rates, and under what defenses or deduplication) and disentangle the novelty of the proposed leakage from the prerequisite capability.

4. The underlying question is important: if a model is trained on curated public data, where the curation used a private target set, does the deployed model leak membership about that private set? However, the current presentation makes the answer hard to find. A concise figure (or two) that lays out the threat model for all three stages—attacker view, defender assets, and leakage channel—would greatly improve readability. With a clearly defined problem and threat model, the empirical results would be far more persuasive; without that foundation, even sophisticated methods risk failing to convince readers.

**Questions:**

Please address the “Weaknesses” above in your rebuttal—especially by (1) stating the problem crisply, (2) explaining why it matters in realistic deployments, and (3) laying out clear threat models for all three attack stages. If those pieces are clarified and make sense, I’m inclined to raise my score.

---

> ### Author Response · Authors · 2025-11-21
> **New Section on Threat Model and Expanded Motivation**
>
> \> **Please spell out—on one page—(i) the adversary’s goal (membership in the private target set used for curation vs. classic training-set membership), (ii) what the adversary can observe at each stage (scores, selection mask, final model), and (iii) what the adversary can do (e.g., can they inject public items?)**
>
> We appreciate the suggestion to improve clarity and added a section on the \**Threat Model\** into Section 3 of the paper. To summarize, the adversary’s goal is to infer the membership of samples in the private target set used to curate a public dataset. This goal is the same at every stage. In all stages, the adversary has knowledge of the public pool dataset. Since it is most likely a web-scale dataset from the internet, we assume the adversary can obtain it. The adversary also knows the candidate samples whose membership they want to infer. This is a standard assumption for membership inference. The adversary also knows the curation algorithm used, which is usually open-source and documented in the model card. Depending on the curation stage, additional knowledge and capabilities change:
>
> 1. Scores: The adversary can observe all scores on the public dataset.
> 2. Subsets: The adversary has access to the subset of public samples with the highest curation scores, with no information of their scores or relative order.
> 3. Models: The adversary only has black-box query access to the model trained on the curated public data subset (with no further knowledge, e.g., the training dataset size). Only in this setting do we assume the adversary can inject a few fingerprint samples into the public data, which Carlini et al. (2024) have shown to be realistic.
>
> \> **The paper does not convincingly explain why leakage at the scoring and subset-selection stages matters when, in many realistic deployments, only the final model is exposed.**
>
> We agree that the motivation can be explained better. We have extended the motivation in the introduction where we outline the following aspects: Curated datasets are often released to the public \[1-3\]. In some cases, even the quality scores *are* released \[4\]. Furthermore, our work on the leakage of the intermediate data is relevant for real life challenges in dictating organization data access, i.e., who should be allowed to access which parts of the pipeline, which is not trivial to determine \[5\]. There also exists a growing market of data curation as a service, where datasets, scores and subsets are passed around (e.g., Scale AI, Snorkel AI, DatologyAI, Cleanlab). For all of these real-life cases, it is important to have some idea of the privacy implications of what is being passed around. Our work is the first to show the risks for two state-of-the-art curation methods on all relevant threat levels (scores, subsets, models).
>
> \> **The paper should justify the realism and scope of this assumption (e.g., where such insertion is feasible, at what rates, and under what defenses or deduplication) and disentangle the novelty of the proposed leakage from the prerequisite capability.**
>
> We appreciate the reviewer raising this important point. Prior work has shown that an adversary has many avenues to insert data into web-scale training datasets \[6\]. Furthermore, prior work has shown that machine learning can be poisoned and backdoored with few modified samples to misclassify images \[7\]. Our end-to-end attacks build on these insights and show that even fewer samples are required to imprint a measurable signal in trained models that an adversary could exploit for membership inference attacks. We show that just five out of a million samples (a rate of 0,0005%) suffice. We furthermore show that these do not need to be mislabeled, but that we can keep the benign caption and just add additional information for the fingerprinting purpose.
>
> We revised the introduction and discussion of our submission to make this distinction clear.
>
> \> **A concise figure (or two) that lays out the threat model for all three stages—attacker view, defender assets, and leakage channel—would greatly improve readability.**
>
> We are grateful for the suggestion on how to improve the presentation. We have included a table in the new section on the **Threat Model.** We believe this better highlights the differences between stages, e.g., what capabilities and access to does the adversary have at each stage.
>
> We thank the reviewer for their readiness to increase their score and, if the above response in fact addressed the reviewer’s concerns, would greatly appreciate it if they adjust the score accordingly.

---

> ### Author Response · Authors · 2025-11-21
> **References for Our Previous Reply**
>
> **References**:
>
> [1]: Li, Z. et al. LAION-SG: An enhanced large-scale dataset for training complex image-text models with structural annotations. (2024).
>
> [2]: Penedo, Guilherme, et al. "The fineweb datasets: Decanting the web for the finest text data at scale." Advances in Neural Information Processing Systems 37 (2024): 30811-30849.
>
> [3]: Penedo, Guilherme, et al. "The refinedweb dataset for falcon llm: Outperforming curated corpora with web data only." Advances in Neural Information Processing Systems 36 (2023): 79155-79172
>
> [4]: Computer, Together Computer, RedPajama: an open dataset for training large language models. (2023).
>
> [5]: Sushko, Y. et al. Our Approach to Protecting AI Training Data. (2025).
>
> [6]: Carlini, N. et al. Poisoning Web-Scale Training Datasets is Practical. in 2024 IEEE Symposium on Security and Privacy (SP) 407–425 (2024). doi:10.1109/SP54263.2024.00179.
>
> [7]: Carlini, N. & Terzis, A. Poisoning and Backdooring Contrastive Learning. in International Conference on Learning Representations (2022). https://openreview.net/forum?id=iC4UHbQ01Mp

---

> > ### Comment · Reviewer_zhSF · 2025-11-27
> >
> > These explanations make sense to me. Beside it, the most interesting aspect after finding the threat, in my view, is how to estimate per-sample privacy-leakage risk (i.e., each example will have an estimated a risk score). If the threats in your paper truly exist, then rather than only hardening the curation pipeline, it seems more rational to estimate per-sample risk before releasing the curated set and simply replace the high-risk samples. This aligns with recent work that scores privacy risk at the example/interaction level and uses MIA-style signals to prioritize mitigation.
> >
> > By contrast, post-training risk control often leaves you with costly remedies such as unlearning or retraining, so pre-publication screening could be far more practical. As background, LiRA and its variants formalize stronger membership tests by focusing on high-precision regimes and likelihood-ratio signals—useful ingredients for robust per-sample scoring.
> >
> > The paper below may be a helpful addition to your discussion section, as it illustrates a concrete pipeline for per-sample (and per-user) privacy risk scoring in recommendation models:
> >
> > He, Gu, Chen (RecSys’25), “RecPS: Privacy Risk Scoring for Recommender Systems.”
> >
> > Since your responses fully address my questions, I’m inclined to accept the paper and raise my score.

---

> > > ### Author Response · Authors · 2025-12-01
> > > **Removing Vulnerable Samples Increases Attack Success**
> > >
> > > We are grateful for the suggestion and helpful pointer.
> > >
> > > We include **new experiments** on the effect of removing high-risk samples, following Carlini et al. (2022) \[1\]. We identify vulnerable samples using average attack success across runs and remove the 5% most vulnerable samples from the target dataset. For comparison, we also remove 5% of samples at random. We report AUC only on the remaining samples.
> > >
> > > - **Baseline**: No samples removed.
> > > - **Ideal**: AUC computed excluding votes from the most vulnerable samples, but without actually removing them from the target dataset. This represents the best-case outcome if removing those samples does not increase vulnerability of the remaining samples.
> > > - **Remove Vulnerable**: 5% most vulnerable samples are removed.
> > > - **Remove Random**: 5% random samples are removed.
> > >
> > > **Effect of Removing Most Vulnerable vs. Random Samples**
> > >
> > > | Method | Baseline | Ideal (Δ) | Remove Vulnerable (Δ) | Remove Random (Δ) |
> > > | :---- | :---- | :---- | :---- | :---- |
> > > | Image-based | 0.8958 | 0.8845 (-0.0113) | 0.9561 (+0.0603) | 0.9010 (+0.0053) |
> > > | TRAK | 0.9055 | 0.9001 (-0.0053) | 0.9400 (+0.0346) | 0.9439 (+0.0383) |
> > >
> > > For both methods, **removing vulnerable samples *increases* overall attack success**.
> > >
> > > **For Image-based curation,** the 5% most vulnerable target samples are the score-determining nearest neighbors for 31.1% of the pool. Their removal exposes 1.8% of previously shielded targets and increases vulnerability for over 80% of target samples, which are now on average the nearest neighbor for 83 additional pool samples. Removing samples at random increases the attack success only marginally. This demonstrates the *Privacy Onion effect* first identified by Carlini et al. (2022) \[1\], where removing vulnerable samples exposes previously protected samples.
> > >
> > > **For TRAK-based curation,** we observe that removing vulnerable samples increases overall attack success similarly to removing random samples. Removing the most vulnerable samples (AUC 0.9400) reduces attack success marginally more than removing random samples (AUC 0.9439). But our ablations have shown that the attack success is very sensitive to target dataset size (Fig. 34-35 in Appendix E.5), so the reduction in target dataset size outweighs the benefit of removing any samples.
> > >
> > > We have added this analysis to the evaluation section and Appendix D.2.
> > >
> > > We thank the Reviewer for their inclination to accept the paper and remain happy to discuss any further comments or questions.
> > >
> > > \[1\] Carlini, Nicholas, et al. "The privacy onion effect: Memorization is relative." *Advances in Neural Information Processing Systems* 35 (2022): 13263-13276.

---

### Official Review · Reviewer_AAT3 · 2025-11-01

**Soundness:** 3
**Presentation:** 3
**Contribution:** 3
**Rating:** 6
**Confidence:** 3

**Summary:**

This paper investigate the privacy risks of data curation and show effective membership inference attacks on all the major steps of data curation. This includes similarity score evaluation of the data points in the public dataset (which is the easiest), and the construction of the curated public dataset (which is harder), and the training of the ML model on the curated dataset. The paper builds upon popular and well-known membership inference attacks and provided versions that works regardless of the data curation method used. Furthormore, custom attack methods are designed for image similarity score-based curation and TRAK-based curation. Experiments show that while data curation does not train any model on the private dataset, but all its major steps still leak the membership of the private domain dataset, and it is worse for smaller domain dataset. This reveals new privacy risks of data curation.

**Strengths:**

1. The paper reveals new privacy risks of data curation via membership inference attacks.
2. Systematic review of MIAs in all steps in the data curation pipeline, and attacks are proposed and validated for all of them.
3. Existing MIAs work for agnostic with modifications, but the paper also proposes custom attacks are proposed to target concrete data curation methods, including TRAK and image-based data curation.

**Weaknesses:**

1. Choice of parameters are not clear or discussed in the main text.
2. Computational complexity of the attacks is not discussed.
3. Limited experiments for end-to-end model MIAs.

Please see the questions listed in the section below.

**Questions:**

1. Some important parameter and implementation details are not explained. For example, how many shadow models $m$ are used in the attacks? What are the parameters in the data curation procedure and what is the setting of the model training procedure? When training the model, is regularization used? If so, how heavy is that?
2. The paper does not discuss the effects of $m$ on the attack performance. I would like to see the relation between the compute spent by the attacker and the attack effectiveness.
3. For end-to-end model MIAs, all results are when the selected target subset is 1% of the target set. I understand that model training is time consuming, but I would like to see the attack performance for more TPRs.

---

> ### Author Response · Authors · 2025-11-21
> **Clarifications and New Ablations**
>
> \> **Choice of parameters are not clear or discussed in the main text.**
>
> We follow the DataComp small-scale training methodology for all experiments. Models are trained from scratch using ViT-B-32 with a batch size of 1024 and automatic mixed precision (AMP). We use the AdamW optimizer with learning rate 5e-4, $\\beta_1=0.9$, $\\beta_2=0.98$, weight decay 0.2, and gradient clipping at 1.0. The learning rate follows a cosine decay schedule with 500 steps of linear warmup. All models are trained for a fixed budget of 10M samples, with the number of epochs automatically adjusted based on the curated pool size (e.g., 100 epochs for 100k samples, 10 epochs for 1M samples). For membership inference attacks, we train 256 shadow models per configuration unless otherwise specified.
>
> Image-Based curation uses the ViT-L/14 CLIP model pretrained by OpenAI to generate the embeddings. TRAK-based curation obtains gradients on a model trained on the full pool dataset and uses as curation scores for each pool sample the average TRAK score for that pool sample to all target samples.
>
> We clarified these aspects in the experimental setup and Appendix D.
>
> \> **Computational complexity of the attacks is not discussed.**
>
> We added ablations on the attack performance over the number of shadow models as Figure 33 to Appendix E. We also include the following table on CIFAR10 in this response:
>
> **Ablation on the Number of Shadow Models (LiRA)**
>
> | Number of Shadow Models | TPR @ 1% FPR |  |
> | :---- | :---- | :---- |
> |  | Image-Based Curation | TRAK |
> | 128 | 0.1031 ± 0.0017 | 0.0138 ± 0.0054 |
> | 64 | 0.0548 ± 0.0012 | 0.0127 ± 0.0081 |
> | 32 | 0.0267 ± 0.0011 | 0.0110 ± 0.0062 |
> | 16 | 0.0098 ± 0.0006 | 0.0096 ± 0.0064 |
>
> Adding more shadow models increases the success to varying degrees for all datasets. We note that the other proposed attacks, except LiRA, do not require shadow models.
>
> \> **Limited experiments for end-to-end model MIAs.  I would like to see the attack performance for more TPRs.**
>
> We performed additional experiments and added full ROC curves for 36 configurations of datasets and target dataset sizes for the end-to-end attack, showing the full TPR-FPR tradeoff in Figures 19-30 in Appendix E. We furthermore add AUC plots, to enhance comparison with other work that has reported these numbers. These results show that, when the attack is successful, it succeeds across a wide range of the trade-off curve.
>
> If the above responses address the Reviewer's concerns, we would greatly appreciate it if they could adjust their score accordingly.

---

> > ### Comment · Reviewer_AAT3 · 2025-11-22
> >
> > I hereby thank the authors for their response and clarification. I will keep my positive score.

---

> > > ### Author Response · Authors · 2025-11-26
> > >
> > > We are grateful for the Reviewer's engagement and their positive assessment of our work. Shall any further questions arise, we remain happy to answer.

---

### Official Review · Reviewer_Ju1r · 2025-11-03

**Soundness:** 2
**Presentation:** 2
**Contribution:** 3
**Rating:** 2
**Confidence:** 3

**Summary:**

This paper presents the first comprehensive privacy analysis of data curation pipelines in machine learning. The authors challenge the common assumption that training models on curated public data inherently protects private target datasets used for curation. Through systematic membership inference attacks across three pipeline stages (curation scores, selected subsets, and final models), the paper demonstrates that each stage leaks information about the private target set.

**Strengths:**

- This is the first systematic analysis of privacy leakage in data curation pipelines, addressing a genuine blind spot in the community. it systematically evaluates privacy leakage across three critical stages of the curation pipeline (curation scores, selected data subsets, and the final trained models) using diverse datasets and curation methods.

- The analysis of influence sparsity in image-based curation provides valuable insights into why certain methods are more vulnerable.

- The paper delivers actionable insights: simple max-based curation is fundamentally insecure, while average-based curation (like TRAK) provides innate robustness except in the small-dataset regime .

**Weaknesses:**

- The threat model is not clearly defined. The adversary's assumed capabilities and knowledge are not clearly stated upfront and appear to change depending on the attack. The adversary knowledge could be escalates to extreme, white-box levels. For the end-to-end TRAK attack, the adversary is assumed to have profound knowledge of the curation mechanism, including the model architecture, the ability to compute gradients, and the need to calculate the Gram matrix $G$. This level of white-box access is not clearly stated or justified.

- The paper's core attack methodology is its adaptation of LiRA. This adaptation is (a) confusingly explained and (b) methodologically questionable. Running an attack algorithm on subsets of the private data to build in/out distributions  is not standard and is not justified as a sound method for simulating the true 'in' and 'out' worlds.

- The paper's strongest end-to-end attack is weakened by its reliance on a proxy metric. The authors state the full end-to-end attack is 'computationally intractable'. Their solution is to 'assume that the adversary can measure whether $f \in \tilde{D}$.' This assumption effectively equates selection with detection, completely ignoring the possibility that ML training dynamics (e.g., SGD noise, and the influence of thousands of other samples) could 'drown out' the fingerprint’s signal, making it undetectable in the final model even if it had been selected.

**Questions:**

- Can you please provide a single, clear definition of the threat model?

- Can you please provide a more detailed justification for your 'shadow' methodology? Specifically, why is re-running the curation algorithm on random subsets of the private data $\mathcal{T}$  a sound way to construct the in/out distributions for a LiRA attack? This seems to be a custom algorithm that borrows the LiRA name, and its statistical validity is not obvious.

- Can you provide any evidence that this assumption of selection equals detection holds? For example, can you show that a fingerprint sample, when selected, always creates a detectable signal in the final model, even when trained as part of a large curated set?

- Could you please analyze the query budget/cost or computational cost of the attack? How would it affect the scalability of the method?

---

> ### Author Response · Authors · 2025-11-21
>
> > **Can you please provide a single, clear definition of the threat model?**
>
> We added a section on the *Threat Model* to Section 3 in the paper. To summarize, the adversary’s goal is to infer the membership of samples in the private target set used to curate a public dataset. This goal is the same at every stage. In all stages, the adversary has knowledge of the public pool dataset. Since it is most likely a web-scale dataset from the internet, we assume the adversary can obtain it. The adversary also knows the candidate samples whose membership they want to infer. This is a standard assumption for membership inference attacks. The adversary also knows the curation algorithm used, which is usually open-source and documented in the model card. Depending on the curation stage, additional knowledge and capabilities change:
>
> 1. Scores: The adversary can observe all scores on the public dataset.
> 2. Subsets: The adversary has access to the subset of public samples with the highest curation scores, with no information of their scores or relative order.
> 3. Models: The adversary only has black-box query access to the model trained on the curated public data subset (with no further knowledge, e.g., the training dataset size). Only in this setting do we assume the adversary can inject a few fingerprint samples into the public data, which Carlini et al. (2024) have shown to be realistic.
>
> > **Can you please provide a more detailed justification for your 'shadow' methodology? Specifically, why is re-running the curation algorithm on random subsets of the private data $\\mathcal{T}$  a sound way to construct the in/out distributions for a LiRA attack?**
>
> We added a subsection dedicated to our adaptation of LiRA to Appendix B.1, explaining why it works and showing a figure of the resulting in and out distributions of the scores. To summarize, the distributions of the observed scores form two shifted normal distributions. We furthermore clarified that these subsets are sampled in such a way, that each target is in exactly half of the subsets. That ensures that our estimates of the in and out distributions are unbiased.
>
> > **This seems to be a custom algorithm that borrows the LiRA name**
>
> Our LiRA-based attack retains key components of the original attack:
>
> - We model in and out distributions using shadow runs on random subsets, parallel to the shadow models of LiRA.
> - We perform per-sample calibration on these in and out distributions.
> - We compute the membership inference scores using the same likelihood ratio.
>
> The adaption is what we are performing inference on, not in the attack methodology itself. Jagielski et al. (2023) similarly adapted LiRA to perform MIA on a teacher training dataset through queries on student models while keeping the name. We believe this accurately describes the statistical methodology and properly attributes Carlini et al.’s (2022) methodological attribution while avoiding false claims of novelty.
>
> > **The authors state the full end-to-end attack is 'computationally intractable'.**
>
> With our statement that “training large models from scratch for **every** membership inference experiment is computationally intractable”, we referred to the fact that **evaluating** the attack on trained models, i.e., training a model from scratch for every membership inference experiment (as would be the most proper evaluation), is prohibitively expensive. As we evaluate six different datasets, two curation methods, multiple seeds, curation parameters, and target dataset sizes, this would add up to over 20,000 GPUh or roughly $75,000 (on the Google Cloud Platform).  We present in the next comment results that show the detection is possible
>
> > **Could you please analyze the query budget/cost or computational cost of the attack? How would it affect the scalability of the method?**
>
> **Computation Cost.** Our proposed attacks are training-free and require running only the respective curation algorithms or operations of similar complexity, therefore scaling linearly with the number of pool samples, as do the curation methods.
>
> - Score-based attacks require one curation run.
> - Subset-based attacks require \~10 curation runs.
> - End-to-end attacks require one curation run.
>
> Our attacks therefore scale significantly better than traditional shadow model-based approaches, which require training many models. This makes our attacks practical at any scale where curation is practical in the first place.
>
> **Query Budget.** To attack a single target, we need at most five black-box queries, one for each fingerprint. As some fingerprints contribute to multiple targets, this scales at most linearly with the number of targets and on expectation sublinear.
>
> We detail this in a new section in Appendix C.4.

---

> ### Author Response · Authors · 2025-11-21
> **New Results Regarding the End-to-End Detectability**
>
> > **Can you provide any evidence that this assumption of selection equals detection holds? For example, can you show that a fingerprint sample, when selected, always creates a detectable signal in the final model, even when trained as part of a large curated set?**
>
> To address the reviewer’s question, we performed additional experiments on detecting the fingerprints. To recall, to attack image-based curation we use mislabeled images (e.g., a photo of an airplane with the caption “*ratatouille*”). For TRAK-based curation we add orthogonal information to benign captions (e.g., a photo of an airplane with the caption “an image of an airplane *and ratatouille*”). We include five of those modified samples in the pool data. This is inspired by the finding from Carlini et al. (2024) \[3\] that this capability is practical and on the finding from Carlini and Terzis (2022) \[4\] that few samples suffice to backdoor a model. We then measure the log-probability of the trained model for predicting the wrong concept (e.g., predicting “*ratatouille*” as the caption for an image of an airplane). We include Figures 10 and 11 with the results in Appendix C and present here a summary table.
>
> **Ablation on the Training Dataset Size for the Fingerprint Detectability**
>
> |  | Log Likelihood-Ratio (Log(P\_Fingerprinted)-Log(P\_Baseline)) |  |
> | :---- | :---- | :---- |
> | Training Dataset Size (\#Fingerprints \= 5\) | Image-Based Curation(mislabeled fingerprints) | TRAK-Based Curation(additional information) |
> |  |  |  |
> | 50,000 | 1.3 | 35.5 |
> | 100,000 | 2.3 | 30.9 |
> | 250,000 | 1.4 | 5.6 |
> | 500,000 | 8.2 | 10.8 |
> | 1,000,000 | 23.5 | 18.8 |
>
> The results  show that across all dataset sizes we evaluated (up to 1,000,000) the fingerprinted models predict orders of magnitude larger probabilities on the fingerprinting concept than baseline models.
>
> We note that for mislabeled fingerprints this difference even increases with training dataset size, because baselines models grow more confident that the mislabeled captions are wrong, while fingerprinted models remain at a nearly constant, much higher, confidence for the wrong label.
>
> If the above responses address the Reviewer's concerns, we would greatly appreciate it if they could adjust their score accordingly.
>
> **References**
>
> \[1\] Jagielski, Matthew, et al. "Students parrot their teachers: Membership inference on model distillation." Advances in Neural Information Processing Systems 36 (2023): 44382-44397.
>
> \[2\] Carlini, Nicholas, et al. "Membership inference attacks from first principles." 2022 IEEE symposium on security and privacy (SP). IEEE, 2022\.
>
> \[3\]: Carlini, N. *et al.* Poisoning Web-Scale Training Datasets is Practical. in *2024 IEEE Symposium on Security and Privacy (SP)* 407–425 (2024). doi:[10.1109/SP54263.2024.00179](https://doi.org/10.1109/SP54263.2024.00179).
>
> \[4\]: Carlini, N. & Terzis, A. Poisoning and Backdooring Contrastive Learning. in *International Conference on Learning Representations* (2022). https://openreview.net/forum?id=iC4UHbQ01Mp

---

> > ### Author Response · Authors · 2025-11-27
> > **Follow-Up**
> >
> > We kindly follow up regarding the rebuttal and check whether our responses address the Reviewer’s concerns.
> >
> > In particular, we have:
> >
> > * Added a dedicated Threat Model section (Section 3.1) with a clear definition of adversary goals and capabilities
> > * Provided detailed justification for our LiRA adaptation in Appendix B.1
> > * Included new experiments (Figures 10-11) demonstrating that fingerprints remain detectable even in models trained on up to 1M samples
> > * Added computational and query cost analysis in Appendix C.4
> >
> > We remain happy to discuss any questions or clarify any points further. We thank the Reviewer again for their time and valuable feedback.

---

### Author Response · Authors · 2025-11-21
**We Thank the Reviewers for the Feedback, Summary of Changes**

We are grateful to all reviewers for their constructive feedback, which greatly helped us  improve our submission. We thank the reviewers for noting our work as “addressing a genuine blind spot in the community” (Reviewer Ju1r), “a new angle on privacy leakage” (Reviewer hymw) and revealing “new privacy risks of data curation” (Reviewer AAT3). We are further happy that they recognize our work as “well motivated and clearly defined” (Reviewer hymw) and “interesting” (Reviewer zhSF) and our empirical evaluation as “convincing and well-presented” (Reviewer hymw) and delivering “valuable” and "actionable insights” (Reviewer Ju1r). We hope that our work can contribute to the community and inform about a “novel and conceptually important” (Reviewer hymw) aspect of privacy leakage.

Below, we present additions to the paper and highlights of the rebuttal that we believe are valuable to share with all reviewers. Following this, we will provide individual responses addressing each Reviewer's comments in detail. A revised version of the paper is uploaded at the same time.

- **We clarified the threat model** and motivated it further based on real examples of released curation scores, released curated datasets, curation as a service, and data access challenges. We added these aspects to the introduction and include a new Section 3.1 dedicated to clarifying the adversary goal and capabilities, and how they vary for attacks on each curation stage.

- **We added a new attack for TRAK scores** based on formulating the membership inference problem as a least squares regression. We show that an adversary can recover membership information for larger target datasets (\~10,000 samples) than shown in the initial submission.

- **We show the robustness of our end-to-end leakage findings** by including additional experiments on the detectability of fingerprints in the training data. We detail how those measurements can be obtained and ablate over the training dataset sizes, showing that the inclusion of fingerprints leads to measurable signals. This holds for both types of fingerprints we employed, i.e., images with *wrong* captions (e.g., an image of an airplane with the caption “*ratatouille*”) in Image-Based curation and images with benign captions with *added wrong information* (e.g., an image of an airplane with the caption “an image of an airplane *and ratatouille*”) for TRAK.

- **We extend the paper to include defenses and leverage and evaluate mitigation strategies based on Differential Privacy.** We evaluate attack performance for various strengths of privacy guarantees. We find that these drastically reduce the attack efficacy, supporting the conclusion of our paper, that curation should incorporate protective measures.

- **We add additional ablations**. We show that both for attacking Image- and TRAK-Based curation, there is a sweet spot where adding or removing dimensions reduces attack success. We show that for LiRA-style attacks, adding more shadow models helps to varying degrees for the different datasets. We ablate over target dataset sizes, showing that the nearest neighbor mechanism of image-based curation consistently exposes samples across all sizes. TRAK-based curation, with its’ averaging, exposes *all* samples at small sizes, but the averaging equally makes it hard to attack *any* sample for large dataset sizes.

---

### Author Response · Authors · 2025-12-03
**Special Summary**

All four reviewers recognized that this work addresses a **novel and important problem**: privacy leakage through data curation pipelines. This attack surface has not been previously studied. Reviewer concerns focused on clarifications and experimental scope, all of which we have comprehensively addressed.

**We conducted a comprehensive revision of the paper.** During the rebuttal period, we added **3 new main-text sections** (Sections 3.1, 4.4, 5), **11 new appendix subsections** (B.1, B.5, C.1, C.4, C.5, D.1, D.2, E.4–E.7) with supporting experiments, and **27 new figures** (Figures 6–13, 19–37). These changes substantially strengthen the paper’s clarity and empirical foundation. Key improvements have been made to address:

- **Motivation** (Introduction): We discuss using real examples (released datasets, released curation scores, curation-as-a-service) the relevance of our work.
- **Threat Modeling** (Sections 3.1): We added a dedicated threat model with a capability table clarifying adversary goals and access at each pipeline stage.
- **End-to-End Attack Validation** (Appendix C.2–C.3)**:** New experiments demonstrate that fingerprints remain detectable in models trained on up to 1M samples—directly addressing concerns that training dynamics might "drown out" the signal. For mislabeled fingerprints, detectability *increases* with dataset size as baseline models grow more confident the wrong label is incorrect.
- **Defenses** (Section 5, Appendix D)**:** We introduce and evaluate DP-based mitigation mechanisms. Key finding: replacing nearest-neighbor scoring with DP-mean reduces TPR to near-random even at ε=1000, while ε=10 effectively neutralizes all attacks. We also show a surprising *Privacy Onion effect* \[1\]: naïvely removing vulnerable samples *increases* overall attack success by exposing previously shielded samples.
- **Comprehensive Ablations** (Section 4.4, Appendix E)**:** We added experiments across embedding dimensions (showing a "sweet spot" for attack success), shadow model counts, target dataset sizes, and 36 full ROC curve configurations. These ablations reveal that image-based curation leaks consistently across scales, while TRAK's averaging provides protection only at larger dataset sizes.
- **Computational Analysis (Appendix C.4):** Our attacks are training-free and require at most 10 curation runs (for subset attacks) and 5 black-box model queries per target (for end-to-end attacks), scaling far better than traditional shadow-model approaches.

We believe these comprehensive updates go beyond fully addressing the concerns raised.

---

To facilitate the Area Chair's independent assessment in the unprecedented absence of further reviewer discussion, we provide the following overview of concerns and their corresponding resolutions (highlighted in blue in the revision).

|Reviewer|Concern / Question|Resolution|Location|
|:----|:----|:----|:----|
|Ju1r|Threat model unclear|Dedicated section with capability table|Sec. 3.1, Table 1|
|Ju1r|LiRA adaptation unjustified|Methodology \+ in/out distribution visualization|App. B.1|
|Ju1r|Selection=detection unvalidated|Fingerprints detectable up to 1M training samples|App. C.2–C.3, Fig. 11–12|
|Ju1r|Computational cost missing, scalability concern|Cost and query budget analysis, scales with curation methods|App. C.4|
|AAT3|Parameters unexplained|Complete training/curation details|Sec. 4, App. C.1, D|
|AAT3|Shadow model ablation|Attack robustness across shadow counts|App. E.6, Fig. 36|
|AAT3|Limited end-to-end TPR evaluation|36 full ROC curves \+ AUC plots|App. E.4, Fig. 20–31|
|zhSF|Threat model unclear / unrealistic|Dedicated section with capability table and justifications based on prior work|Sec. 3.1, Tab. 1|
|zhSF|Motivation for intermediate attacks|Real-world examples (released datasets and curation scores, curation-as-a-service)|Sec. 1|
|zhSF|Injection realism|Discussion referencing Carlini et al. (2024) \[2\]|Sec. 1, Sec. 3.1|
|zhSF|Per-sample risk scoring|Additional experiments on removing vulnerable samples|App. D.2|
|hymw|Motivation for intermediate attacks|Real-world examples (released datasets and curation scores, curation-as-a-service)|Sec. 1|
|hymw|No defense mechanisms|DP mitigations (ε=2 to 1000\) and sample removal evaluated|Sec. 5, Fig. 6, App. D, Tab. 2-3|
|hymw|Missing ablations|Embedding dim, dataset size ablations|App. E, Fig. 20–37|

We thank the AC for their consideration given the exceptional circumstances.

---

**References**

\[1\] Carlini, Nicholas, et al. "The privacy onion effect: Memorization is relative." *Advances in Neural Information Processing Systems* 35 (2022): 13263-13276.

\[2\]: Carlini, N. *et al.* Poisoning Web-Scale Training Datasets is Practical. in *2024 IEEE Symposium on Security and Privacy (SP)* 407–425 (2024). doi:[10.1109/SP54263.2024.00179](https://doi.org/10.1109/SP54263.2024.00179).

---

### Meta-Review · Area_Chair_VCci · 2025-12-19

**Summary:**

The paper demonstrates that data curation, which selects samples from a large public collection based on private data, and consequently training an ML model on the selected samples can expose the hidden private data to MIA.

The reviewers note the following weaknesses:
1. Unclear threat model definition.
2. LiRA methodology questionable.
3. Only proxy demonstration of the end-to-end attack against model training.
4. Parameter choices not described sufficiently.
5. Computational complexity is not discussed.
6. Limited results for end-to-end attacks.
7. Lack of justification why leakage at intermediate steps is important.
8. End-to-end attack weakened by poisoning assumption.
9. Limited discussion on mitigations.
10. No ablations on dataset size and embedding dimension.

**Reviewer Concerns:**

The authors have provided an extensive response and revision that addresses reviewers' concerns.

Specifically:
1. Added table clarifying the threat model. Resolved.
2. Explained the methodology more clearly. Resolved.
3. Provided some additional results and indication that full resolution would be quite costly. It is not clear why the authors could not run the experiments in a selected setting to keep the cost more manageable. Partially resolved.
4. Description added. Resolved.
5. Added ablation on number of shadow models to show performance degradation with less compute. Concrete numbers of how much compute each attack used would still have been a nice addition. Partially resolved.
6. Added more end-to-end attack results. Resolved.
7. Clarified the justification. Resolved.
8. Clarified the justification of why the assumption is realistic under some conditions. Resolved.
9. Added more mitigation results with differential privacy. Resolved.
10. Added the requested ablations. Resolved.

Overall, most reviewer comments appear to have been resolved, which the exception of #3 and #5 which are only partially resolved.

**Reviewer Scores:**

The scores of the original reviews were 6,6,2,2.

Reviewer zhSF with score 2 was able to participate in the discussion to report changing their mind to accept as their concerns were addressed. I agree with the assessment that their concerns were addressed.

Reviewer Ju1r with score 2 did not participate in the discussion. Their concerns were partially addressed by the clarification of the threat model and the attack methodology. However, the concern about lack of end-to-end results was addressed only partially. I assume the reviewer would have raised their score, but it is unclear whether they would have recommended acceptance.

From my own evaluation of the paper, I find it to be well executed and written and to provide an important finding on the vulnerability of data curation. The results on vulnerability of intermediate datasets and curation scores are already important, so the lack of complete end-to-end results is not critical for the value of the paper.

In light of these, I recommend the paper to be accepted.

Further suggestions:
The abbreviation TRAK is not explained in the Introduction. Please explain it.

---

### Decision · Program_Chairs · 2026-01-26

Accept (Poster)